# Cell-type-specific functionality encoded within the intrinsically disordered regions of OCT4

Burak Ozkan [1,2,8], Mitzy Rios de Anda[1,2,8], Elisa Hall-Ponsele[1,2], Maria Rosa Portero Migueles[1], Amani Alshaikh[1,2,3], Marta Hanzevacki[1,2], Moriyah Naama[4], Katharine Furlong[1,2], Gareth A. Roberts [1,2], Meryam Beniazza[1], My Linh Huynh[1], Michael R. O'Dwyer[1,2], Sonia Yiakoumi[1], Christos Spanos [5], Hazar Yassen[4], Keisuke Kaji [1], Hitoshi Niwa[6,7], Yosef Buganim [4], Sally Lowell [1,2] & Abdenour Soufi [1,2] ✉

The cell-type-specific function of transcription factors (TFs) is crucial for determining cellular identity. However, it is unclear how a single TF can function specifically in different cell types. Here, we define the molecular features that enable OCT4 to reprogram somatic cells into pluripotent or trophoblast stem cells, maintain the self-renewal of embryonic stem cells (ESCs), and drive lineage commitment during early embryonic development. Embedded within the intrinsically disordered regions (IDRs) of OCT4, we uncover short linear peptides that are essential for reprogramming (SLiPERs) but dispensable for ESC self-renewal. SLiPERs adopt a quasi-ordered state and, during reprogramming, recruit a unique set of proteins to closed chromatin that are unnecessary for ESC self-renewal. Interestingly, SLiPERs are essential for embryos to develop beyond late gastrulation. Removing SLiPERs leads to aberrant OCT4 binding, derailing the regular transition of ESCs out of pluripotency. Our findings identify modules within IDRs that contribute to the functional versatility and specificity of TFs.

Cell identity is defined by the functions of transcription factors (TFs) that are specific to each cell type and can operate differently depending on the cellular context[1,2]. For example, OCT4, in combination with SOX2, KLF4 and c-MYC (OSKM) play a crucial role in establishing and maintaining pluripotency in early embryonic development and embryonic stem cells (ESCs) but can also induce pluripotent stem cells (iPSCs) from fully differentiated cells[3–10]. Furthermore, OSKM can also induce other unrelated lineages including neuronal cells and extra-embryonic trophoblasts[11–14]. In

another combination, OCT4 can cooperate with GATA3, KLF4 and c-MYC (OGKM) to induce trophoblast stem cells (hiTSCs) from human fibroblasts[15], supporting a different role of OCT4 in establishing the trophectoderm fate in human[16,17]. OCT4 can also be combined with neuronal TFs to convert differentiated cells into neurons[18]. Overall, OCT4 plays a multifaceted role ranging from establishing and maintaining pluripotency to promoting pluripotency exit, lineage specification and establishing the germ line[5,16,19–29]. This raises an important question of whether the diverse

[1]Institute of Regeneration and Repair, Centre for Regenerative Medicine, University of Edinburgh, Edinburgh, UK. [2]Institute of Stem Cell Research, School of Biological Sciences, University of Edinburgh, Edinburgh, UK. [3]King Abdulaziz City for Science and Technology Health Sector, Riyadh, Saudi Arabia. [4]Department of Developmental Biology and Cancer Research, Institute for Medical Research Israel-Canada, The Hebrew University-Hadassah Medical School, Jerusalem, Israel. [5]Wellcome Discovery Research Platform for Hidden Cell Biology, Michael Swann Building, Edinburgh, UK. [6]Institute of Molecular Embryology and Genetics, Kumamoto University, Kumamoto, Japan. [7]RIKEN Center for Developmental Biology, Kobe, Japan. [8]These authors contributed equally: Burak Ozkan, Mitzy Rios de Anda. ✉e-mail: Abdenour.Soufi@ed.ac.uk

functions of OCT4 stem from a common mechanism or several separable modes of action.

TFs are modular proteins that contain functionally independent domains[30]. They recognise specific DNA sequences through DNA-binding domains (DBDs) and interact with partner proteins via effector domains (EDs)[30–33]. OCT4 contains a bi-partite POU-DBD composed of a POU-specific domain (POU_S) and a POU-homeodomain (POU_{HD})[34,35], which are connected by a linker and flanked by two EDs; an N-terminal (NTD) and a C-terminal (CTD). While OCT4-DBD has been directly implicated in reprogramming and pluripotency maintenance by directly interacting with DNA, nucleosomes and other protein partners[35–44], the functional contribution of OCT4-EDs remains poorly defined. For example, OCT4 has been reported to interact with both co-activators and co-repressors to maintain pluripotency[45–50]. Yet, the gene activation function of OCT4 seems sufficient for maintaining pluripotency[51]. Recent evidence suggests that many TF-EDs including OCT4-EDs are intrinsically disordered regions (IDRs) and exert their function by mediating liquid-liquid phase separation (LLPS) to form protein condensates on chromatin essential for gene transactivation[52–55]. Hitherto, OCT4-LLPS has been considered functionally generic, suggesting that any TF-ED with low-complexity amino acid composition can fulfil this role[37,51,52]. Since the knowledge of protein motifs or folds is often insufficient to infer function[56], it is challenging to predict if OCT4-EDs exert any specific function.

In this work, we hypothesised that the various functions of OCT4 during reprogramming into iPSCs and iTSCs, pluripotency maintenance, and lineage commitment could be dissected apart from each other at subdomain resolution. We have therefore systematically dissected OCT4 to show that defined modules within OCT4-IDRs contribute to the selective and diverse functions of OCT4.

## Results

### Short linear peptides within OCT4 are required for reprogramming

To define the functional segments of OCT4 that are required for reprogramming to iPSCs, at the sub-domain resolution, we used a library of OCT4 mutants ($n = 119$), each containing a stretch of five amino acids (a.a.) deletion (del) tiling across the full-length of the protein with two a.a. overlap (Fig. 1a)[44]. Using a dox-inducible lentiviral system, we screened for the ability of each OCT4 deletion (del) in combination with SKM to reprogram mouse embryonic fibroblasts (MEFs) to iPSCs, which are positive for the pluripotency marker Nanog as detected by immunofluorescence (Fig. 1b, c). As expected, no NANOG-positive iPS colonies were generated in the absence of OCT4, confirming that OCT4 is essential for reprogramming (Fig. 1c, d)[3]. Furthermore, del-47 to del-70 and del-79 to del-95, which overlap with POU_S and POU_{HD} within OCT4-DBD, respectively, abolished the reprogramming activity of OCT4, demonstrating the importance of OCT4 DNA- and nucleosome-binding activities (Fig. 1d)[44]. We also generated equivalent alanine stretch OCT4 mutants (ala), which all showed similar reprogramming efficiency and expression levels (Supplementary Fig. 1a–c). Interestingly, all deletions within the linker region (del-71 to del-78) impaired OCT4 reprogramming activity apart from del-72 and the equivalent alanine stretch (ala-72) mutation, that had minimum effect on reprogramming (Fig. 1d and Supplementary Fig. 1a, b). Residues within del-72 are the only region within the linker that can adopt a defined structure (Supplementary Fig. 1d)[35]. We therefore isolated three independent iPSC lines using each of OCT4-WT, del-72 and ala-72 mutants that were genotyped by sequencing to confirm the presence of the expected mutations (Supplementary Fig. 1e, f). The resulting iPSC lines can differentiate to all three germ layers from embryoid bodies, confirming their pluripotency (Supplementary Fig. 1g, h). Thus, the unstructured part of the linker is more essential for reprogramming compared to the structured part.

Interestingly, sequential deletions (3 or more) appearing as defined patches within OCT4-NTD and -CTD substantially impaired the reprogramming capacity of OCT4 (shaded in Fig. 1d). We refer to these Short Linear Polypeptides Essential for Reprogramming as SLiPERs, which were separated by other segments (nonSLiPERs) that showed minimum or even positive effects on reprogramming when deleted from OCT4 (Fig. 1d). Hence, OCT4 reprogramming activity is encoded within defined regions of the unstructured NTD, CTD, the linker, and the structured DBD.

### OCT4 SLiPERs are dispensable for ESC self-renewal

To assess whether OCT4 uses the same domains for reprogramming as ES self-renewal, we measured the ability of the OCT4 mutants to rescue self-renewal in a conditionally null *Pou5f1* ESC line (ZHBTc4.1)[6,37]. These cells had both endogenous alleles of *Pou5f1* inactivated and contained a dox-off *Pou5f1* transgene, so they can be propagated in an undifferentiated state in the absence of dox and undergo differentiation upon dox treatment (Supplementary Fig. 2a). However, in the presence of dox, ZHBTc4.1 self-renewal can be rescued by ectopically expressing *Pou5f1* (Fig. 1e, f). Like reprogramming, OCT4 mutants containing deletions within POU_S and POU_{HD} failed to rescue ZHBTc4.1 self-renewal, highlighting the importance of OCT4-DBD in both pluripotency induction and maintenance (Fig. 1g). However, del-72 and ala-72 mutations within the structured portion of the linker had negative effects on self-renewal, suggesting more important role in ESC self-renewal than reprogramming (Fig. 1g and Supplementary Fig. 2b–d). More strikingly, deletions across OCT4-NTD and -CTD had little effect on OCT4 ability to maintain pluripotency, supporting their functional redundancy[37]. Two exceptions were OCT4-del-104 and del-113 within OCT4-CTD, which abolished ESC self-renewal (Fig. 1g). Moreover, ala-stretch OCT4 mutants showed similar ES self-renewal capacity to their deletion counterparts (Supplementary Fig. 2b–d). Notably, OCT4 functionality in reprogramming and ESC self-renewal showed no significant correlation with amino acid conservation (Fig. 1h). Overall, many OCT4-SLiPERs appear dispensable for ESC self-renewal, suggesting different contributions to inducing and maintaining pluripotency.

To define what constitutes functional SLiPERs, we examined the cumulative effects of merging the sequential 5 a.a. deletions on reprogramming and ESC self-renewal (Fig. 1i). We deleted (Δ mutants) or replaced with a linker (lin. mutants) three SPLiPERs (a.a. 5-15, 29-42, 311-321) and three nonSLiPERs (a.a. 95-117, 125-138, 341-354) either individually or in combination (Fig. 1i). We also established that the mRNA and protein levels of these OCT4 mutants are comparable to wildtype, except for OCT4-Δ-mini showing low protein levels (Supplementary Fig. 3a–c). OCT4-SLiPER mutants displayed diminished reprogramming function without significantly affecting ESC self-renewal, confirming that SLiPERs are uniquely essential for reprogramming (Fig. 1j, k). However, most OCT4 nonSLiPERs mutants displayed no change and sometimes enhanced reprogramming without affecting ES self-renewal (Fig. 1j, k). The one exception is the non-SLiPERs (a.a.125-138), which directly precedes OCT4-DBD, when deleted alone or in combination with other regions (Δ125-138 and Δ-mini) impaired both reprogramming and ESC self-renewal, but not when replaced with a linker (lin125-138, lin-mini, and mini) (Fig. 1j, k). Thus, although a.a.125-138 sequence is not required for pluripotency, a generic linker is required for OCT4 functionality. OCT4-lin-mini, and -mini (missing three nonSLiPERs) were the only mutants that reduced ESC self-renewal but not reprogramming (Fig. 1j, k). In summary, OCT4-SLiPERs constitute defined reprogramming modules within OCT4-IDRs.

### Pluripotent and trophoblast stem cells are induced by different OCT4 domains

To investigate whether OCT4-SLiPERs display similar functions in human and mouse reprogramming, we tested one SLiPER mutant

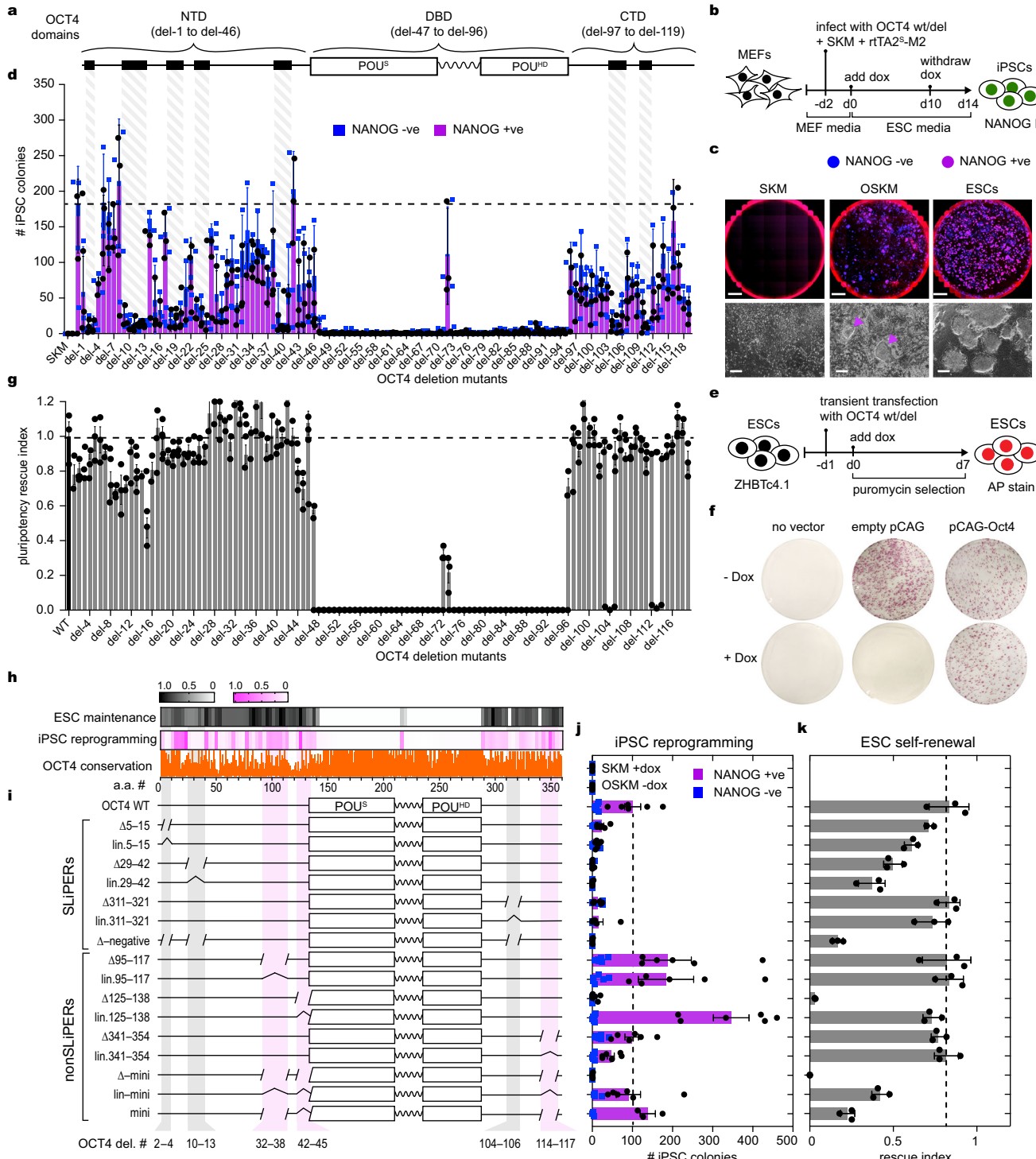

**Fig. 1 | OCT4 employs distinct domains to induce and maintain pluripotency.**
**a** Schematic of OCT4 domains including POU^S and POU^HD DBDs (white boxes) separated by linker (zigzag line), and the NTD and CTD (black lines). The OCT4 deletion mutations are indicated above. Black boxes indicate the mapped SLiPERs highlighted in (**d**). **b** Experimental scheme for reprogramming MEFs to iPSCs. **c** Whole-well images (top panels) showing DAPI fluorescence (blue), NANOG immuno-fluorescence (red) and merged (magenta) of iPSCs. SKM and ESCs were used as negative and positive controls, respectively. Bright-field images (bottom panels) of typical iPSC colonies (purple arrows). Images are representative of *n* = 3 biological replicates. Scalebars 5 mm (top) 100 μm (bottom). **d** Bar plots of NANOG positive (purple) versus NANOG negative (blue) colonies for each OCT4 deletion mutant in combination with SKM. SKM +Dox and OSKM -/+ Dox (bars 1–3) were used as controls. The dotted line indicates the average reprogramming activity of

OCT4 WT. **e** Experimental scheme for pluripotency rescue assay in ZHBTc4.1 ESCs upon dox-treatment to repress *Pou5f1* transgene with examples shown in (**f**). **g** Bar plots showing the ability of OCT4 deletion mutants to maintain ZHBTc4.1 ESCs self-renewal expressed as a rescue index, which is the ratio of AP-stained colonies in +Dox over -Dox. OCT4 WT was used as a control (dashed line). **h** Heatmaps showing the reprogramming and pluripotency maintenance activities from (**d**) and (**g**), respectively. OCT4 protein conservation is shown (orange). **i** Schematics showing OCT4 mutants lacking SLiPERs (grey) and nonSLiPERs (magenta). **j** Bar plots of NANOG positive (magenta) and NANOG negative (blue) iPSC colonies generated by OCT4 WT and mutants shown in (**i**). **k** Bar plots of pluripotency rescue index of OCT4 WT and mutants shown in (**i**). Data are presented as mean values +/- SEM from *n* = 3 (**d**, **g** and **k**) and *n* = 5 (**j**) independent biological replicates. Source data are provided as a Source Data file.

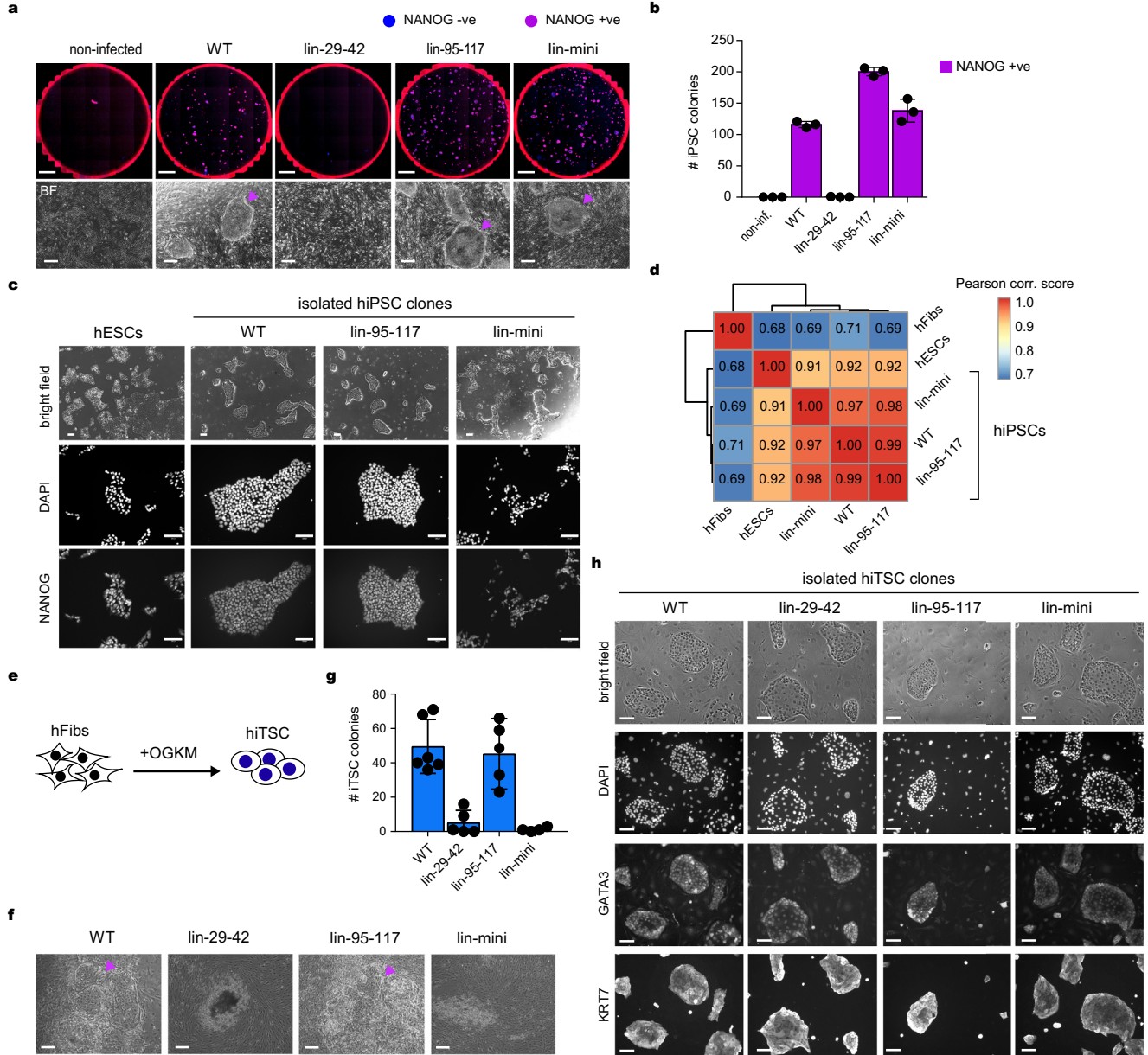

**Fig. 2 | Inducing pluripotent and trophoblast stem cells requires functionally distinct OCT4 domains. a** Whole-well images showing DAPI (blue), NANOG immuno-fluorescence (red) and merged (magenta) after reprogramming human fibroblasts to iPSCs using OCT4 WT or mutants (scalebar, 5 mm). Non-infected fibroblasts were used as controls. Bright-field images (bottom panels) of typical iPSC colonies (purple arrows). Scalebar, 100 μm. **b** Bar plot of NANOG positive human iPSC colonies (purple) generated by OCT4 WT and mutants. **c** Immunofluorescence images showing the expression of the pluripotency marker NANOG in hiPSCs generated using OCT4 WT and mutants compared to hESCs. DAPI staining and bright-field images are also shown. Scalebars indicate 100 μm. **d** Pearson correlation of RNA-seq between human fibroblasts, iPSCs generated by OCT4 WT and mutants and hESCs. The heatmap scale is indicated. **e** Experimental scheme for reprogramming human fibroblasts to iTSCs. **f** Bright-field images showing typical morphology of iTSC colonies obtained using OCT4 WT or mutants. Scalebar, 100 μm. **g** Bar plot of human iTSCs generated by OCT4 WT and mutants, which were scored based on morphology. **h** Immunofluorescence images showing the expression of human TSC markers GATA3 and KRT7 in hiTSCs generated using OCT4 WT and mutants. DAPI staining and bright-field images are also shown. Scalebar, 100 μm. Images are representative of n = 3 biological replicates (**a**, **c**, **f** and **h**). Data are presented as mean values +/- SEM from n = 3 (**b**) and n = 5 (**g**) independent biological replicates. Source data are provided as a Source Data file.

(lin29-42) and one nonSLiPER mutant (lin95-117) individually, and three nonSLiPERs mutations together (lin-mini) in reprogramming human adult fibroblasts (hFibs) to hiPSCs. Indeed, OCT4 nonSLiPER mutants displayed enhanced reprogramming compared to OCT4 WT, unlike the SLiPER mutant, which was reprogramming-deficient (Fig. 2a, b). The resulting hiPSCs from OCT4-lin29-42 and lin-mini mutants as confirmed by Sanger sequencing (Supplementary Fig. 4a, b), expressed the pluripotency marker NANOG and showed similar gene expression pattern to hESCs as measured by RNA-seq, indicating full

reprogramming (Fig. 2c, d, and Supplementary Fig. 4c). The pluripotency of these iPSC lines was further confirmed by their ability to form embryoid bodies and differentiate spontaneously in vitro to all three germ layers (Supplementary Fig. 4d, e). Altogether, OCT4-SLiPERs serve similar roles in reprogramming mouse and human fibroblasts to pluripotency.

OCT4 along with GATA3, KLF4 and c-MYC (OGKM) were shown to induce trophoblast stem cells (hiTSCs) from human fibroblasts independently of pluripotency (Fig. 2e)[15]. Thus, we were curious to know

whether inducing TSCs requires similar OCT4 SLiPERs to iPSCs. To that end, we performed reprogramming of hFibs into hiTSCs using OCT4 WT, the SLiPER mutant lin29-42 and the nonSLiPER mutants lin95-117 and lin-mini (Fig. 2f). Like iPSCs, OCT4-lin29-42 mutant was detrimental in reprograming to iTSCs, indicating that this region of OCT4 is crucial for both reprograming processes (Fig. 2g, and Supplementary Fig. 4f). Interestingly, removing one nonSLiPER (lin95-117) caused no significant effect on hiTSC reprogramming as seen in hiPSCs, while removing three nonSLiPERs in OCT4-lin-mini mutant showed diminished ability to induce hiTSCs, contrary to hiPSC reprogramming (Fig. 2f, g and Supplementary Fig. 4f). Nevertheless, all hiTSC lines generated by OCT4 mutants displayed typical morphology and expressed key TSC markers (Fig. 2h). Altogether, these data indicate that inducing pluripotency and trophoblast stem cells can be dissected apart by removing defined OCT4 regions.

## OCT4-SLiPERs exert specific functions independent of LLPS

Both SLiPERs and nonSLiPERs are located within OCT4-IDRs, displaying similar disorder propensity (blue line in Fig. 3a). OCT4-IDRs are known to drive the formation of phase-separated condensates that can be observed as localized puncta in the nuclei[57]. To investigate whether SLiPERs and nonSLiPERs affect the phase-separating capacity of OCT4 during reprogramming, we examined the formation of OCT4 nuclear puncta after ectopically inducing OCT4-WT, lin29-42, lin95-117 and lin-mini mutants along with SKM in human fibroblasts for 48 h (OSKM-48h). High-resolution confocal microscopy revealed that OCT4-WT and mutants all formed nuclear puncta during early reprogramming, without any apparent difference (Fig. 3b and Supplementary Fig. 5a). Thus, OCT4-SLiPER mutations block reprogramming without disrupting LLPS.

Next, we examined whether OCT4-SLiPER mutants impede reprogramming by impairing the transactivation function of OCT4, which was shown to be important for inducing and maintaining pluripotency[37,51,58]. We therefore measured the ability of OCT4-WT, lin29-42, lin95-117 and lin-mini mutants to drive the expression of a luciferase reporter from the *FGF4* and *UTF1* enhancers (Fig. 3c)[6,40]. Interestingly, the reprogramming-deficient OCT4-lin29-42 mutant displayed strong luciferase transactivation equivalent to OCT4-WT and lin95-117 mutant (Fig. 3c). The reprogramming-efficient OCT4-lin-mini mutant, however, showed reduced luciferase transactivation (Fig. 3c). Thus, the specific function exerted by OCT4-SLiPERs in reprogramming does not correlate with generic transactivation function[6]. Furthermore, we investigated whether fusing the strong transactivation domain of VP16 to OCT4 can substitute OCT4-SLiPERs in reprogramming. However, instead of enhancing reprogramming, VP16 reduced the ability of OCT4 to induce pluripotency (Supplementary Fig. 5b, c), which is consistent with the negative effects of strong transactivation on reprogramming[51]. Importantly, lin29-42-VP16 fusion remained deficient in reprogramming like the non-fusion counterpart (Supplementary Fig. 5b, c), confirming that OCT4-SLiPERs cannot be replaced by any transactivation domains in reprogramming. Altogether, these data demonstrate that OCT4-SLiPERs serve a specific function in reprogramming beyond mediating generic transactivation through LLPS.

It is noteworthy that IDRs can contain molecular recognition features (MoRFs), which are short regions that permit interaction with structured proteins and undergo disorder-to-order transitions (induced folding), indicative of their functionality[59]. Indeed, the MoRF prediction algorithm MoRFpred has identified that SLiPERs including a.a. 29-42 are more likely to contain MoRFs compared to nonSLiPERs such as a.a. 95-117 (red line in Fig. 3a)[60]. Interestingly, AlphaFold2 algorithm predicts that OCT4-SLiPERs such as a.a. 29-42 adopt a "quasi-ordered" state with low to very low predicted local difference distance test (pLDDT) scores (Fig. 3d). Thus, SLiPERs may be unstructured in isolation but can be induced to fold, unlike the fully-unstructured nonSLiPERs[61]. Moreover, a protein-protein interaction domain located within residues 29-42 is highly probable as calculated

by ANCHOR2 (Supplementary Fig. 5d)[62]. In summary, OCT4-SLiPERS may fold upon interactions with specific protein partners to mediate reprogramming.

## Removing SLiPERs drive aberrant OCT4 binding during reprogramming

We have previously shown that OCT4 interaction with nucleosomes is essential for reprogramming[43,44,63]. To investigate whether OCT4-SLiPERs are involved in nucleosome binding, we have generated OCT4 WT, lin29-42, lin95-117 and lin-mini mutants as recombinant proteins (Supplementary Fig. 5e), and measured their affinity for both naked DNA and nucleosomes, using electrophoretic mobility shift assays (EMSA)[43]. All OCT4 derivatives displayed similar affinity for naked DNA and nucleosomes, demonstrating comparable pioneer activity (top panels in Fig. 3e). Furthermore, the affinity of OCT4-WT and both mutants to naked DNA and nucleosomes was abolished in the presence of excessive amounts of specific DNA competitors but not non-specific competitors, indicating similar DNA-binding specificity (bottom panels in Fig. 3e). Thus, removing SLiPERs or nonSLiPERs is inconsequential to the specific interaction of OCT4 with nucleosomes.

Next, we examined whether SLiPERs mediate OCT4 binding with other protein partners on chromatin using chromatin immunoprecipitation combined with selective isolation of chromatin associated proteins (ChIP-SICAP), which simultaneously maps genome occupancy and chromatin-bound protein interactors (Fig. 3f)[46]. We carried out ChIP-SICAP on OCT4 WT, lin95-117, lin29-42 and lin-mini mutants during the early stages of reprogramming hFibs (OSKM-48h) as well as in hESCs to examine whether the proteins that colocalise with OCT4 to maintain pluripotency are also required for reprogramming. We confirmed that protein levels and chromatin fragmentation were similar across all samples for appropriate comparisons (Supplementary Fig. 5f, g). First, we identified the genomic locations enriched for OCT4 WT and mutants, confirming that OCT4 initial engagement with the somatic genome is markedly different from that in pluripotency, as reported previously (Supplementary Fig. 5h)[63–65]. In early reprogramming, a significant number of genomic sites ($n = 81,866$) bound by OCT4-WT were also enriched for the other mutants (Fig. 3g). However, the SLiPER mutant OCT4-lin29-42 was also enriched at unique sites ($n = 49,086$) and excluded from other sites ($n = 13,074$) (Fig. 3g). While the shared and OCT4-lin29-42 excluded sites were largely within closed chromatin as measured by ATAC-seq, the unique sites targeted by OCT4-lin29-42 were mainly within accessible chromatin (Fig. 3g, h). Notably, the open sites targeted exclusively by OCT4 lin29-42 mutant were not enriched for OCT4 motifs, unlike the other sites (Fig. 3g). These open sites targeted by OCT4 lin29-42 mutant were also more proximal to transcription start sites (TSS) of genes mainly associated with stress-response, apoptosis and cellular adhesion genes (Supplementary Fig. 5i, j). Accordingly, the induction of OSKM expression increased the caspase 7/9 activity using all OCT4 variants, but the OCT4-lin29-42 mutant caused significantly more apoptosis compared to the other OCT4 counterparts (Fig. 3i). Altogether, removing SLiPERs drive aberrant OCT4 binding to non-specific sites within open promoters of apoptotic and stress-response genes that may negatively interfere with reprogramming.

## OCT4 SLiPERs recruit a unique set of proteins to chromatin in reprogramming

Next, we used mass-spectroscopy (MS) to identify proteins associated with OCT4 WT, lin29-42, lin95-117 and lin-mini mutants in early reprogramming and colocalise with OCT4 in hESCs. After filtering non-specific proteins that interact with OCT4 antibody in hFibs (in the absence of OCT4) or interact with IgG in hESCs, the identified proteins from two biological replicates showed significant overlap (Supplementary Fig. 6a). Gene ontology analysis confirmed that the identified OCT4 partners were predominantly associated with chromatin-based

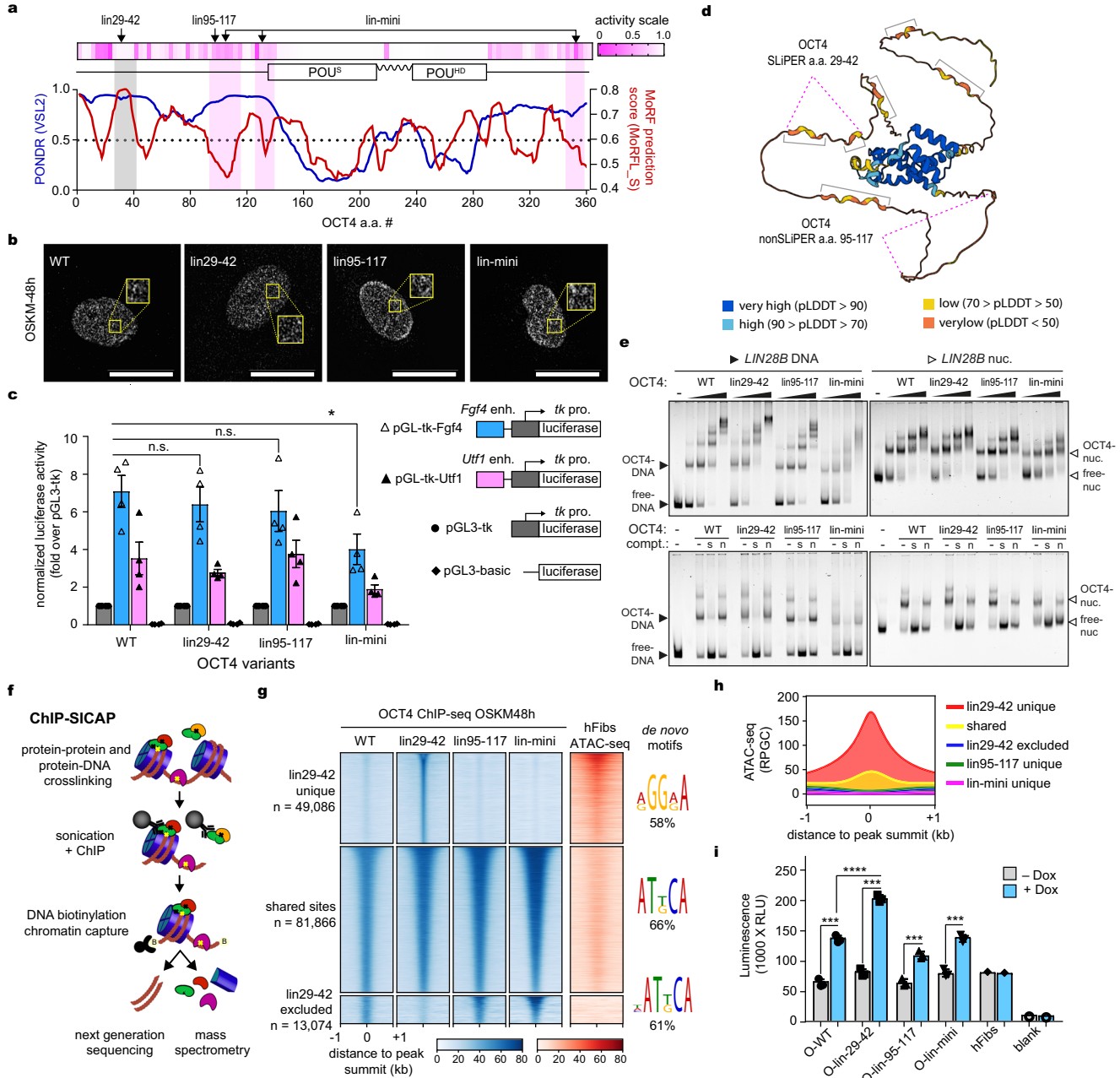

**Fig. 3 | Removing SLiPERs drives aberrant OCT4 binding to accessible chromatin during reprogramming. a** Graph of intrinsic disorder of OCT4 by PONDR-VSL2 (blue) and MoRF prediction by MoRFpred (red). Heatmap of OCT4 reprogramming activity and OCT4 domains are indicated above. SLiPERs and non-SLiPERs are highlighted in grey and magenta, respectively. **b** Immunofluorescence showing punctate staining of OCT4-WT and mutants in OSKM-48h (insets of zoomed-in views in yellow). Scalebar, 20 μm. **c** The transactivation of Luciferase reporter constructs (schematics) by OCT4-WT and mutants normalized against Renilla luciferase and expressed as a ratio over empty vector (pGL3-tk). **d** Cartoon representation of OCT4 structure by AlphaFold2 colour-coded by pLDDT scores. Examples of SLiPERs and nonSLiPERs are indicated. **e** Representative EMSA of increasing amounts (0–8 nM) of recombinant OCT4-WT and mutants binding to *LIN28B*-DNA (top-left panel) and *LIN28B*-nucleosome (top-right panel) at 1 nM concentration. The affinity of recombinant OCT4 WT and mutants (3 nM) for *LIN28B*-DNA (bottom-left panel) and *LIN28B*-nucleosome (bottom-right panel) in the presence of 30-fold molar excess (60 nM) of specific competitor ("s" lanes) or

non-specific competitor ("n" lanes) or absence of competitor ("-" lanes). **f** ChIP-SICAP experimental workflow for identifying OCT4 partners on chromatin. Dual crosslinking with formaldehyde and glutaraldehyde to preserve protein-DNA and protein-protein interactions, respectively. **g** ChIP-seq read density heatmaps (blue) of OCT4 WT and mutants in OSKM-48h relative to chromatin accessibility (red) spanning ±1 kb from the centre of OCT4 peaks. The sequences were rank-ordered according to ChIP-seq read density. The number of OCT4 sites (*n*) and the colour scales are indicated. Motifs enriched in OCT4 sites are shown. **h** Profile plots showing the average ATAC-seq signal around OCT4 sites shown in (**g**). **i** Bar plots showing the caspase activity to indicate the level of apoptosis induced by OCT4 wild type and mutants in OSKM-48h. Data are presented as mean values +/- SEM of *n* = 4 (**c**) and *n* = 3 (**i**) biological replicates. Statistical significance assessed by unpaired *t*-test as indicated by *P* values (*≤ 0.05, ***≤ 0.001, ****≤ 0.0001) (**c** and **i**). Images are representative of *n* = 6 (**b**) and *n* = 3 (**e**) biological replicates. Source data are provided as a Source Data file.

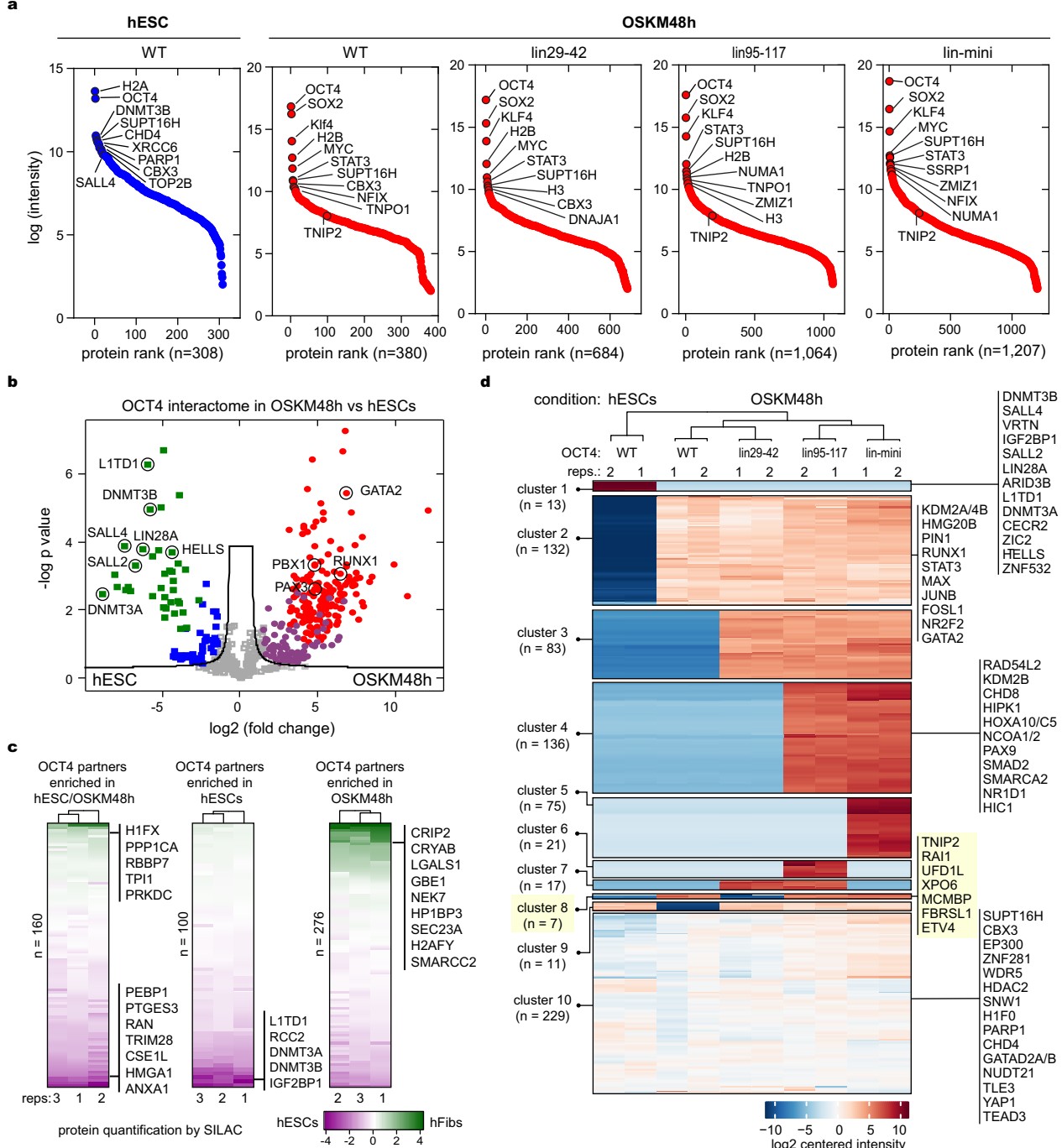

**Fig. 4 | OCT4 engages a unique set of proteins on chromatin during reprogramming. a** Proteins associated with OCT4 WT or mutants on chromatin identified by ChIP-SICAP rank ordered by normalized signal intensity from two biological replicates ($n=2$). Top ranked proteins and TNIP2 are highlighted. **b** Volcano plot illustrating the differentially enriched proteins co-localised with OCT4 in early reprogramming versus ESCs. Statistical significance assessed by unpaired $t$-test and FDR-adjusted $P$ value. The $-\log10$ ($P$ value) is plotted against the log2 fold change. The significance threshold is represented by the non-axial lines denoting; |log2(fold change)| > 1.5 and FDR = 0.1. OCT4 partners enriched uniquely in ESCs (green squares), shared but more enriched in ESCs (blue squares), enriched uniquely in early reprogramming (red circles), shared but more enriched in early reprogramming (purple circles), or shared and equally enriched (grey squares). Example proteins are indicated. **c** Heatmaps showing the differential abundance of OCT4-interacting proteins as measured by SILAC-MS and expressed as a log2 hFibs/hESCs ratio (colour intensity scale at the bottom) from $n=3$ biological replicates. Examples of OCT4 partners and number of proteins ($n$) are indicated. **d** Heatmap showing hierarchal clustering of proteins identified by ChIP-SICAP for OCT4 WT and mutant in hESCs and early reprogramming. Examples from selected groups are highlighted. Number of proteins ($n$) identified in each group is indicated. The colour intensity scale is shown at the bottom. Source data are provided as a Source Data file.

processes (Supplementary Fig. 6b). Furthermore, histones, TFs and chromatin modifiers were amongst the top-ranked proteins associated with OCT4 during early reprogramming and in ESCs, confirming their colocalization on chromatin (Fig. 4a).

Principal component analysis (PCA) showed that proteins associated with OCT4 WT and mutants in early reprogramming form a distinct cluster from those that colocalise with OCT4 in hESCs, which are also different from non-specific interactors (Supplementary

Fig. 6c). Direct comparison of OSKM-48h and hESCs showed a markedly different set of chromatin-associated proteins co-localizing with OCT4 (Fig. 4b). For example, the somatic TFs GATA2, PBX1, PAX3 and RUNX1 are exclusively enriched during early reprogramming, supporting the notion that OCT4 can redistribute these factors during reprogramming (Fig. 4b)[64,66–68]. Functional protein association network analysis (STRING) also identified proteins involved in cell death to be uniquely enriched during reprogramming, indicating that OCT4 may induce apoptosis in reprogramming by directly interacting with apoptotic proteins as well as targeting apoptotic genes (Supplementary Fig. 6d)[63]. Moreover, pluripotency network proteins such as L1TD1, LIN28A, DNMT3A/B, HELLS were identified as unique OCT4 partners in hESCs, which were also previously identified in mouse ESCs (Fig. 4b and Supplementary Fig. 6d)[46,69]. Nevertheless, many proteins in the OCT4 interactome are shared between early reprogramming and hESCs like the chromatin modifiers CBX3, CHD4, WDR5, KDM1A and SUPT16H (Fig. 4a and Supplementary Fig. 6d). To investigate whether the disparity in OCT4 interactomes is due to protein abundance, we quantified the amounts of these proteins using stable isotope labelling with amino acids in cell culture (SILAC) proteome analysis (Supplementary Fig. 6e–g). Interestingly, apart from a minor set of proteins, that are somatic- or pluripotency-specific, the majority of the differentially-enriched OCT4 partners in early reprogramming or in hESCs are equally abundant in both cell types (Fig. 4c). Additionally, OCT4-associated proteins that are equally enriched in reprogramming and hESCs show similar variation in abundance (Fig. 4c). Therefore, protein abundance is not the main reason for the distinct OCT4 engagement with the proteome, supporting specific interactions.

Next, we compared the interactomes of OCT4 WT, lin95-117, and lin29-42 mutants in early reprogramming and hESCs. We only considered proteins that are consistent in both replicates across all samples ($n = 724$). Using unbiased hierarchical clustering, we identified 7 protein clusters (Fig. 4d). Surprisingly, OCT4-lin29-42 mutant, despite the lack of SLiPER, can still interact with most proteins associated with OCT4-WT in reprogramming and in hESCs. This reprogramming deficient mutant gains more interactions that are also gained by the other reprogramming-active lin95-117, lin-mini mutant compared to OCT4 WT (cluster 3, $n = 83$ in Fig. 4d). Therefore, these OCT4 protein partners are not sufficient for inducing pluripotency. However, many proteins interact exclusively with reprogramming-efficient OCT4 lin95-117 and lin-mini mutants (cluster 4, $n = 136$ in Fig. 4d), which have been previously shown to enhance reprogramming, such as RAD54, KDM2B, CHD8, SMAD2, SMARCA2, and PAX6[48,70–73]. This may therefore account for the positive effect of removing the nonSLiPERs on reprogramming.

To identify protein partners mediated by OCT4-SLiPERs that are required for inducing but not maintaining pluripotency, we identified a set of 7 proteins (cluster 8) that interact with OCT4 WT, lin95-117, and lin-mini mutants (reprogramming-efficient) but not with OCT4-lin29-42 mutants (reprogramming-deficient) and not in hESCs (yellow box in Fig. 4d). Apart from the ETS Variant Transcription Factor 4 (ETV4), which has been previously shown to enhance reprogramming[74], the remaining proteins in cluster 8 proteins are yet to be linked to reprogramming. These include TNIP2 (TNFAIP3-interacting protein), RAI1 (Retinoic acid-induced protein), UFD1L (Ubiquitin recognition factor in ER-associated degradation protein), XPO6 (Exportin-6), MCMBP (Minichromosome maintenance complex-binding protein), and FBRSL1 (Fibrosin-1-like protein). Taken together, these data show that OCT4-SLiPERs may promote interactions with proteins that are specifically required for reprogramming.

## TNIP2 rescues the reprogramming activity of OCT4-SLiPER mutant

To examine the functional contribution of OCT4-interactors in cluster 8 during reprogramming, we used reprogrammable-MEFs (cas9-TNG-MKOS-MEFS) containing a constitutive Cas9 expression construct into the Rosa26 locus, a Nanog-GFP-IRES-Puro knock-in reporter, and Dox-inducible MKOS-IRES-mOrange polycistrionic reprogramming cassette in the *Sp3* locus (Fig. 5a, b)[73]. Infecting cas9-TNG-MKOS-MEFS with sgRNA lentiviruses targeting all the genes encoding for cluster 8 proteins significantly decreased the reprogramming efficiency as measured by GFP-positive colonies (Nanog +ve) compared to control genes known to be essential for reprogramming, such as *Kdm4b* and *Stat3* (Fig. 5c, d). Next, we examined the effect of knocking out cluster 8 genes on ESC self-renewal using cas9-TNG-ES cells, which constitutively express Cas9 and a Nanog-GFP reporter (Supplementary Fig. 7a)[75]. Apart from *Ufd1l* and *Mcmbp*, knocking out cluster 8 genes showed no effects on ES self-renewal (Supplementary Fig. 7b, c). We also investigated the effect of knocking out cluster 8 genes on the proliferation of cas9-MEFs by measuring CellTrace™ Far Red dye dilution using flow cytometry (Supplementary Fig. 7d). Again, apart from *Ufd1l* and *Mcmbp*, which abolished MEF proliferation to the same extent as Myc-KO control, MEFs were largely not affected by knocking out the rest of cluster 8 genes (Supplementary Fig. 7e, f). The detrimental effects of *Ufd1l* and *Mcmbp* knockouts in both MEFs and ESCs are consistent with their critical roles in proteostasis and DNA replication, respectively[76,77]. Therefore, apart from *Ufd1l* and *Mcmbp*, which showed the most severe reprogramming phenotype, the rest of the genes in cluster 8 play a specific role in reprogramming, indicating that OCT4-SLiPERs mediate interaction with proteins that are crucial for reprogramming.

Next, we investigated the chromatin localisation of cluster 8 proteins by isolating the cytoplasmic, nuclear (soluble), and the chromatin fractions from hESCs, OSKM-48h expressing OCT4-WT and mutants, and hFibs (Supplementary Fig. 7g). Remarkably, TNIP2 and UFD1L were associated with chromatin in early reprogramming only in the presence of OCT4-WT and the reprogramming-efficient mutants (lin-95-117 and lin-mini) but not in the presence of OCT4-lin29-42 mutant (reprogramming-deficient) in OSKM-48h or hFibs and ESCs (Fig. 5e). XPO6 and ETV4 were also more enriched in chromatin in OSKM-48h compared to hFibs, albeit OCT4-lin29-42 had no impact on this enrichment and similar enrichment was observed in hESCs (Fig. 5e). We were unable to assess the enrichment of RAI1, MCMBP, and FBRSL1 on chromatin due to the low specificity of the available antibodies. This data indicates that OCT4 SLiPERs may recruit TNIP2 and UFD1L to chromatin to specifically induce pluripotency.

Due to the unique role of TNIP2 in reprogramming, we set out to illustrate the mechanism by which OCT4-SLiPERs-TNIP2 interaction mediates reprogramming. TNIP2, also known as A20-binding inhibitor of NF-κB (ABIN-2), is classified as a cytoplasmic zinc-finger protein that functions as a negative regulator of NF-κB activation in response to multiple inflammatory stimuli[78]. However, the truncated TNIP2 C-terminus (TNIP2CT) has been observed to enter the nucleus in transformed cell lines[79]. Hence, we ectopically expressed TNIP2 and TNIP2CT along with OSKM in hFibs and confirmed that only TNIP2CT can readily enter the nucleus in early reprogramming (Fig. 5f). To physically link TNIP2CT to OCT4, we have generated a TNIP2CT-OCT4-lin29-42 fusion protein (TNIP2CT-O$^{lin29-42}$), and showed that TNIP2CT-O$^{lin29-42}$ can also enter the nucleus like OCT4-lin-29-42 during early reprogramming (Fig. 5f). Remarkably, fusing TNIP2CT to OCT4-lin29-42 can effectively rescue OCT4 reprogramming activity unlike co-expressing TNIP2CT with OCT4-lin29-42 which failed to induce pluripotency (Fig. 5g, h). Therefore, the physical interaction between TNIP2CT and OCT4-SLiPERs on chromatin is what mediates reprogramming.

The binding of TNIP2 to linear polyubiquitin plays a crucial role in mediating NF-κB signalling in the cytoplasm[80,81]. To investigate the role of TNIP2-ubiquitin binding in reprogramming, we generated a TNIP2CT ubiquitin binding mutant and fused to OCT4-lin29-42 (TNIP2CT$^{ub}$-O$^{lin-29-42}$)[80]. Interestingly, TNIP2CT$^{ub}$-O$^{lin-29-42}$ displayed

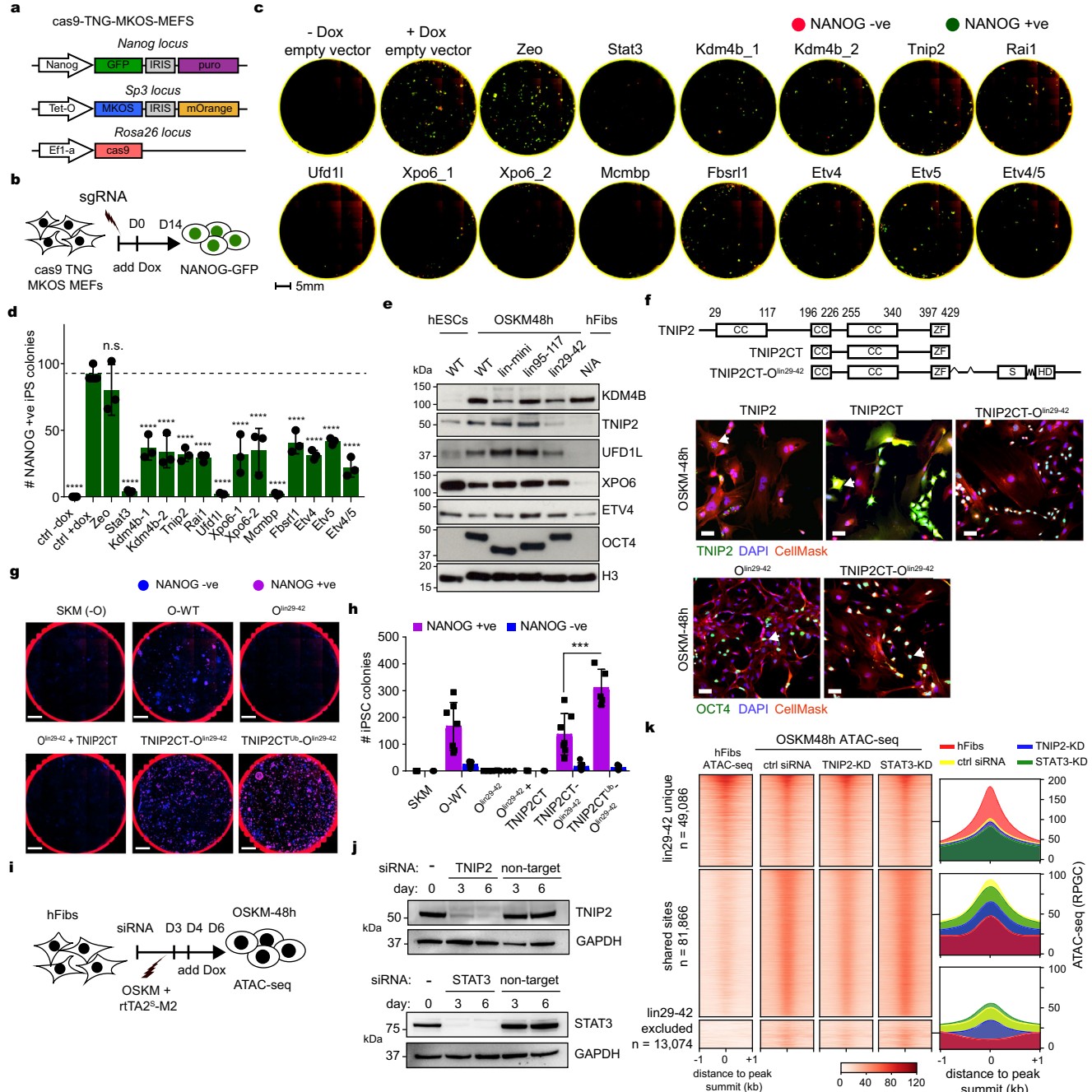

**Fig. 5 | TNIP2 rescues the reprogramming activity of OCT4-SLiPER mutant.**
**a** Schematic of reprogrammable MEFs (cas9-TNG-MKOS-MEFs), expressing GFP under *Nanog* promoter, cas9 under *Elf-a* promoter, and dox-inducible MKOS-ires-mOrange. **b** Experimental scheme for reprogramming cas9-TNG-MKOS-MEFs to iPSCs after targeting specific genes with sgRNA. **c** Whole-well images showing GFP (green) and RFP fluorescence (red) colonies after gRNA targeting (indicated on top). **d** Bar plots of GFP positive iPSC colonies shown in (**c**). **e** Western blots of OCT4 partners in hESCs, OSKM-48h, and hFibs. Molecular weight standards (kDa) are indicated. Histone H3 was used as loading control. **f** Immunofluorescence showing the nuclear localisation (green) of TNIP2CT and TNIP2CT-OCT4 fusion compared to TNIP2 (protein schematics shown above). DAPI (blue) and CellMask (red) staining were used to indicate the nucleus and the plasma membrane, respectively. Scale-bar, 100 μm. **g** Whole-well images showing DAPI (blue), NANOG (red) and merged (magenta) of iPSCs generated by TNIP2CT-OCT4 fusion derivatives compared to OCT4 WT and lin29-42 mutant. Scalebar, 5 mm. **h** Bar plots of NANOG +ve colonies

(purple) versus DAPI (blue) generated by OCT4 derivative shown in (**g**).
**i** Experimental flowchart of ATAC-seq during the early reprogramming after knocking down *TNIP2* using siRNA. *STAT3* and non-targeting siRNA were used as controls. **j** Western blots confirming diminished protein levels of TNIP2 (top) and STAT3 (bottom) after siRNA-KD in day 3 and 6 as shown in (**i**). **k** ATAC-seq read density heatmaps spanning ±1 kb from the centre of OCT4 peaks identified in (Fig. 3g) during early reprogramming and after TNIP2-KD and STAT3-KD as shown in (**i**). Average profile plots are shown to the left. Number (*n*) of OCT4 sites and ATAC enrichment scale (RPGC) are indicated. Images are representatives of *n* = 3 (**c**, **f**, and **j**) and *n* = 6 (**g**) biological replicates. Data are presented as mean values +/- SEM of *n* = 3 (**d**) and *n* = 6 (**h**) biological replicates. Statistical significance assessed by one-way ANOVA (**d**) or unpaired *t*-test (**h**) as indicated by adjusted *P* values (n.s > 0.05, *≤ 0.05, **≤ 0.01, ***≤ 0.001, ****≤ 0.0001). Source data are provided as a Source Data file.

more efficient reprogramming than the WT counterpart (Fig. 5g, h). Thus, binding ubiquitin is not important for TNIP2 function in reprogramming. TNIP2 has also been reported to exert transcriptional activation by recruiting the chromatin-remodelling complex SWI-SNF through direct interaction with BAF60a subunit[79]. We have therefore explored the effects of TNIP2 knockdown on chromatin accessibility during early reprogramming using on-target short interfering RNA (siRNA) (Fig. 5i). We confirmed significant reduction of TNIP2 and STAT3 proteins, which we used as a positive control, compared to non-targeting siRNA (Fig. 5j). Interestingly, TNIP2-KD resulted in less chromatin opening of closed sites targeted by OCT4-WT and mutants (shared sites in Fig. 5k). However, the accessibility of open sites that were exclusively bound by lin29-42 mutant was reduced to similar levels in TNIP2-KD, STAT3-KD, and siRNA control (lin29-42 unique sites in Fig. 5k). Thus, OCT4-SLiPER recruits TNIP2 to closed sites to induce chromatin opening during reprogramming.

### OCT4 SLiPERs are required for embryos to develop through late gestation

Since OCT4 is also important for lineage commitment, we aimed to examine whether OCT4 SLiPERs are required during the transition of pluripotency from the naïve to a primed state in the pre and post-implantation epiblast, respectively, and the transition out of pluripotency at the onset of germ layer differentiation during late gastrulation[82,83]. We have therefore replaced the SLiPER region (a.a.29-42) within the endogenous *PouSF1* gene in the E14tg2a mouse ESC line with a glycine linker using co-delivery of Cas9 ribonucleoprotein (RNP) and single-stranded DNA (ssDNA) for homology-directed repair (HDR) (Fig. 6a)[84]. After genotyping and sequencing, we isolated three independent ESC lines all containing a lin-29-42 insertion in one allele and an in-frame deletion of a.a. 34-42 in the other allele (*PouSF1*<sup>lin/Δ</sup>) (Supplementary Fig. 8a). The isolated *PouSF1*<sup>lin/Δ</sup> ESC lines express comparable amounts of OCT4 to the wildtype counterpart in both LIF and LIF+2i culture conditions (Fig. 6b and Supplementary Fig. 8b). Since the deletion and linker substitution of OCT4 a.a.29-42 were both deficient in reprogramming but maintained ES self-renewal (Fig. 1i–k), we considered these *PouSF1*<sup>lin/Δ</sup> ES lines as appropriate to study the function of this OCT4-SLiPER. After subcutaneous injection of *PouSF1*<sup>lin/Δ</sup> ES lines into nude mice we obtained teratomas like in wildtype ESCs (Supplementary Fig. 8c). Histological examination revealed that these teratomas contained derivatives of all three germ layers, including neural tissues, cartilage, and columnar epithelium (Supplementary Fig. 8d). These data demonstrate that the *PouSF1*<sup>lin/Δ</sup> ES cells exhibit pluripotency.

To further evaluate the pluripotency of *PouSF1*<sup>lin/Δ</sup> ESCs, we measured their contribution levels to embryonic chimaeras at the blastocyst (pre-implantation) and the E6.5 epiblast (post-implantation) stages by aggregating *PouSF1*<sup>lin/Δ</sup> ESCs (GFP +ve) with a morula-stage host embryo (GFP -ve). All three *PouSF1*<sup>lin/Δ</sup> ESC lines showed efficient contribution to the inner cell mass of blastocysts and the post-implantation epiblast at E6.5 (Fig. 6c–e). This contrasts with *PouSf1* knockout embryos (*PouSf1*<sup>-/-</sup>), which usually arrest at the blastocyst stage E5.0[5]. However, *PouSF1*<sup>lin/Δ</sup> embryos began to disintegrate around E8.5 and caused increased foetal resorption during mid-gestation at E13.5 (Fig. 6e–h and Supplementary Fig. 8e, f). This is consistent with the conditional knockout of *PouSF1* shortly after post-implantation, which impeded lineage specification at ~E8[21]. To confirm that *PouSF1*<sup>lin/Δ</sup> ESCs have a defect in lineage commitment, we cultured them in the absence of LIF to trigger differentiation. A significant number of *PouSF1*<sup>lin/Δ</sup> ESCs continued to self-renew when plated at clonal density in the absence of LIF, whereas most of *PouSF1*<sup>wt/wt</sup> ESCs initiated differentiation under the same condition (Fig. 6i, j). Thus, *PouSF1*<sup>lin/Δ</sup> ESCs exhibit mis-regulated differentiation, which is more apparent in chimaeras where there is a more stringent requirement for properly regulated differentiation compared to teratoma assays.

### Removing SLiPERs induces aberrant OCT4 binding in ES cells

Next, we sought to examine whether mutations in OCT4-SLiPERs can cause transcriptional and chromatin changes at the pluripotency stages that may be responsible for mis-regulating ESC differentiation. We have therefore measured gene expression of *PouSF1*<sup>lin/Δ</sup> ESCs using RNA-seq under both LIF+2i and LIF culture conditions, compared to wildtype ESCs. PCA analysis revealed a difference between gene expression of *PouSF1*<sup>lin/Δ</sup> mutant and wildtype ESCs especially when cultured under the LIF+2i condition (Fig. 7a and Supplementary Fig. 9a). Differentially expressed genes (DEGs) included lineage-specific TFs such as Nkx2.9, Hnf4a and Foxd3 under LIF+2i conditions (Fig. 7b). However, this was also associated with ~5-fold upregulation of the pluripotency TF NANOG both at the RNA and protein levels, especially under LIF+2i conditions (Fig. 7b, c and Supplementary Fig. 9b). Thus, the upregulation of NANOG may enhance *PouSF1*<sup>lin/Δ</sup> ESC self-renewal, consistent with previous studies[85].

Next, we measured chromatin accessibility in *PouSF1*<sup>lin/Δ</sup> mutant and wildtype ESCs under LIF conditions using ATAC-seq. Interestingly, there were limited changes in chromatin accessibility in *PouSF1*<sup>lin/Δ</sup> mutant and wildtype ESCs (Fig. 7c, d). We then assessed whether OCT4-lin/Δ mutants and wildtype engage chromatin differently during ESC self-renewal using ChIP-seq. Despite the significant overlap between OCT4 lin/Δ-mutant and wildtype binding sites (common sites), which mainly occurs within accessible chromatin ($n = 27{,}728$ peaks), OCT4-lin/Δ mutants display significantly more enrichment in almost twice as many sites (mutant sites, $n = 52{,}879$ peaks) occurring at less accessible sites (Fig. 7e). OCT4 wildtype is also more enriched at some unique sites albeit less frequently (wildtype sites, $n = 8741$ peaks). Surprisingly, OCT4-lin/Δ mutants were less enriched at the *Nanog* promoter and to a lesser extent at the enhancer compared to wildtype, despite elevated *Nanog* expression (highlighted in yellow in Fig. 7c). Next, we examined whether OCT4-lin/Δ mutants display different motif readout on nucleosomes by first identifying nucleosome bound by OCT4 and then mapping motif enrichment around their dyads[65]. Overall, nucleosomes show similar enrichment and positioning within OCT4 common, mutant, and wildtype sites (Fig. 7f). However, nucleosomes specifically targeted by OCT4 mutant are more enriched for SOX motifs in addition to OCT4 motifs, particularly at the entry and exit, indicating more co-binding with SOX2 or other HMG box TFs (Fig. 7f, g). Strikingly, genes targeted exclusively by OCT4 mutant are associated with a single GO term (chromatin silencing at rDNA), which is not specific to early development or pluripotency, unlike the genes targeted by both OCT4 wildtype and mutant or wildtype only (Fig. 7h). Thus, the mis-regulated differentiation of *PouSF1*<sup>lin/Δ</sup> ESCs may be due to elevated *Nanog* expression and the aberrant OCT4 binding, targeting random genes.

## Discussion

The combinatorial function of TFs underlies the basic principle of cellular identity, and this simple rule has been consistently used to reprogram differentiated cells across all lineages[1,2]. However, almost all lineage-specific TFs are involved in specifying multiple cell types, which poses an important question about their functional specificity. Many models have been proposed to reconcile the dogma about functional plasticity versus specificity including: (1) TF cooperative and competitive DNA binding result in cell-type-specific occupancy of gene enhancers and promoters[33,36,86]. (2) Protein-protein interactions enable TFs to recruit cell-type-specific co-factors to chromatin[87,88]. (3) Extracellular signals establish cell-type-specific TF assemblies on gene regulatory elements[89,90]. All the above interactions occur in synchrony, making it challenging to decipher the essential molecular features that impart TFs' selective and malleable functions.

In this study, we systematically dissected OCT4 to define the molecular basis that elicits its multifaceted function. We discover SLiPERs within OCT4-IDRs to play distinct roles in reprogramming and

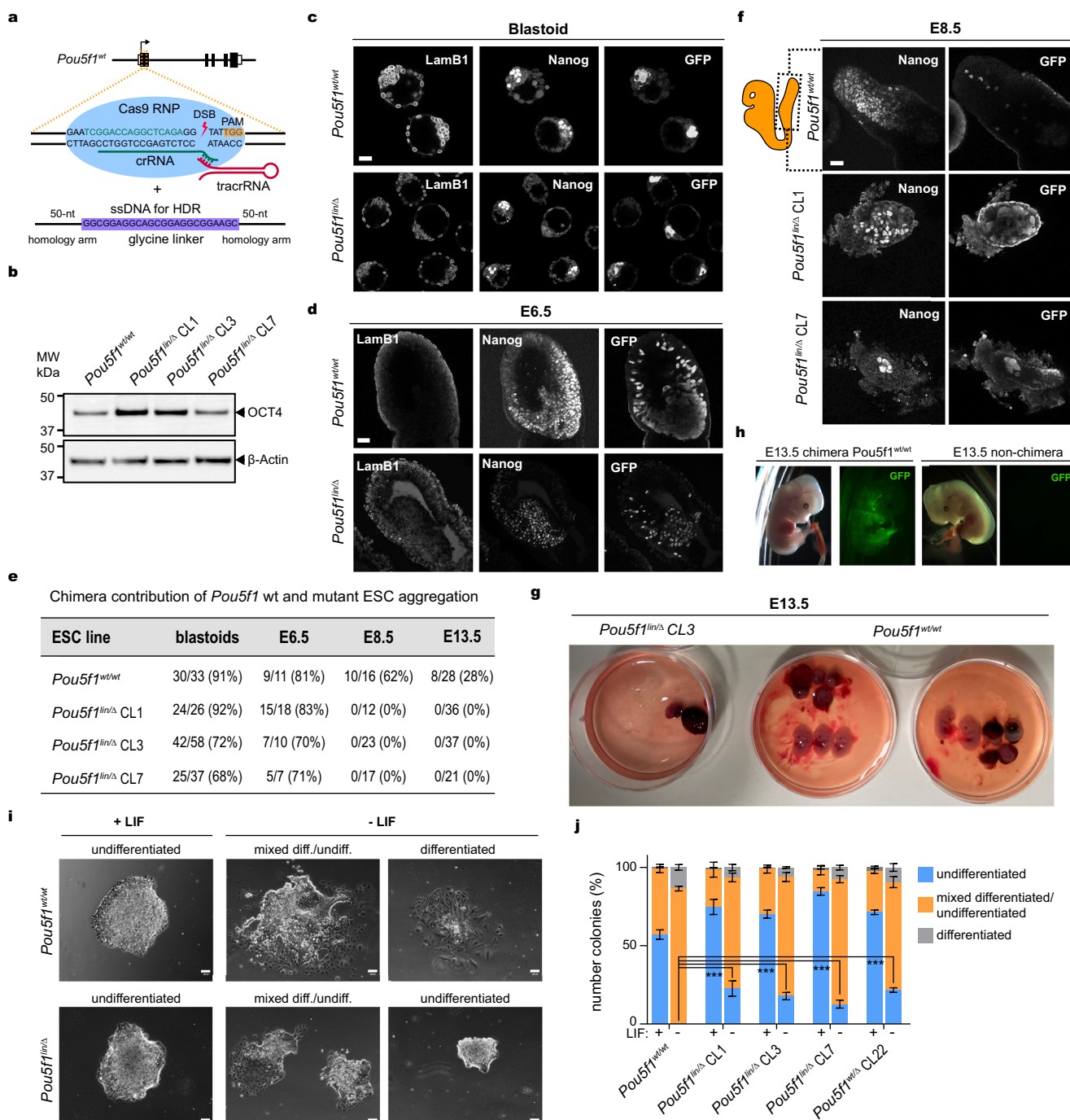

**Fig. 6 | OCT4 SLiPERs are required for embryos to develop beyond late gastrulation. a** Schematic representation of the knock-in strategy used to replace OCT4-SLiPER with a linker in the endogenous *Pou5f1* gene in ES cells. A crRNA was designed to introduce a double-stranded break (DSB) downstream OCT4-SLiPER coding for a.a. 29-42 (green) by Cas9 RNP (blue) bound to PAM (yellow). A single-stranded donor DNA that harbours glycine-rich linker coding sequence (purple) flanked by 50-mer homology arms on both sides was used for homology-directed repair (HDR). **b** Western blot showing the OCT4 levels in three independent ESC clones with *Pou5f1^lin/Δ^* knock-in compared to the E14tg2a parental line (*Pou5f1^wt/wt^*). Molecular weight standards (kDa) are indicated. **c** Immunofluorescence of LaminB1, Nanog, and GFP, showing the contribution of *Pou5f1^lin/Δ^* ES lines to the inner cell mass of blastocysts compared to *Pou5f1^wt/wt^*. Scalebar, 20 μm. **d** Same as (**c**) for E6.5 embryos. **e** Table summarizing the contribution of *Pou5f1^wt/wt^* and

*Pou5f1^lin/Δ^* ESC lines to chimeric embryos at different developmental stages. **f** Same as (**c**) for E8.5 embryos. Inset showing the lateral view representation of embryos. **g** *Pou5f1^lin/Δ^* ESCs caused increased foetal resorption and failed to generate chimeric embryos at E13.5 compared to *Pou5f1^wt/wt^*. **h** The contribution of *Pou5f1^wt/wt^* ESCs to chimeric embryos at E13.5 were assessed by transgene GFP fluorescence. **i** Colony morphology of ESCs in the presence and absence of LIF, showing the robustness of *Pou5f1^lin/Δ^* ESC self-renewal even in the absence of LIF. Scalebar, 100 μm. **j** Quantitation of colony types formed by different *Pou5f1^lin/Δ^* ES lines under different conditions as shown in (**h**). Data are presented as mean values +/- SEM of n = 3 biological replicates. Statistical significance was assessed by multi-comparison one-way ANOVA as indicated by P value (***≤0.001). Images are representatives of n = 3 (**b**, **h** and **i**) and n = 59, 47, 68 (**c**, **d** and **f**) biological replicates. Source data are provided as a Source Data file.

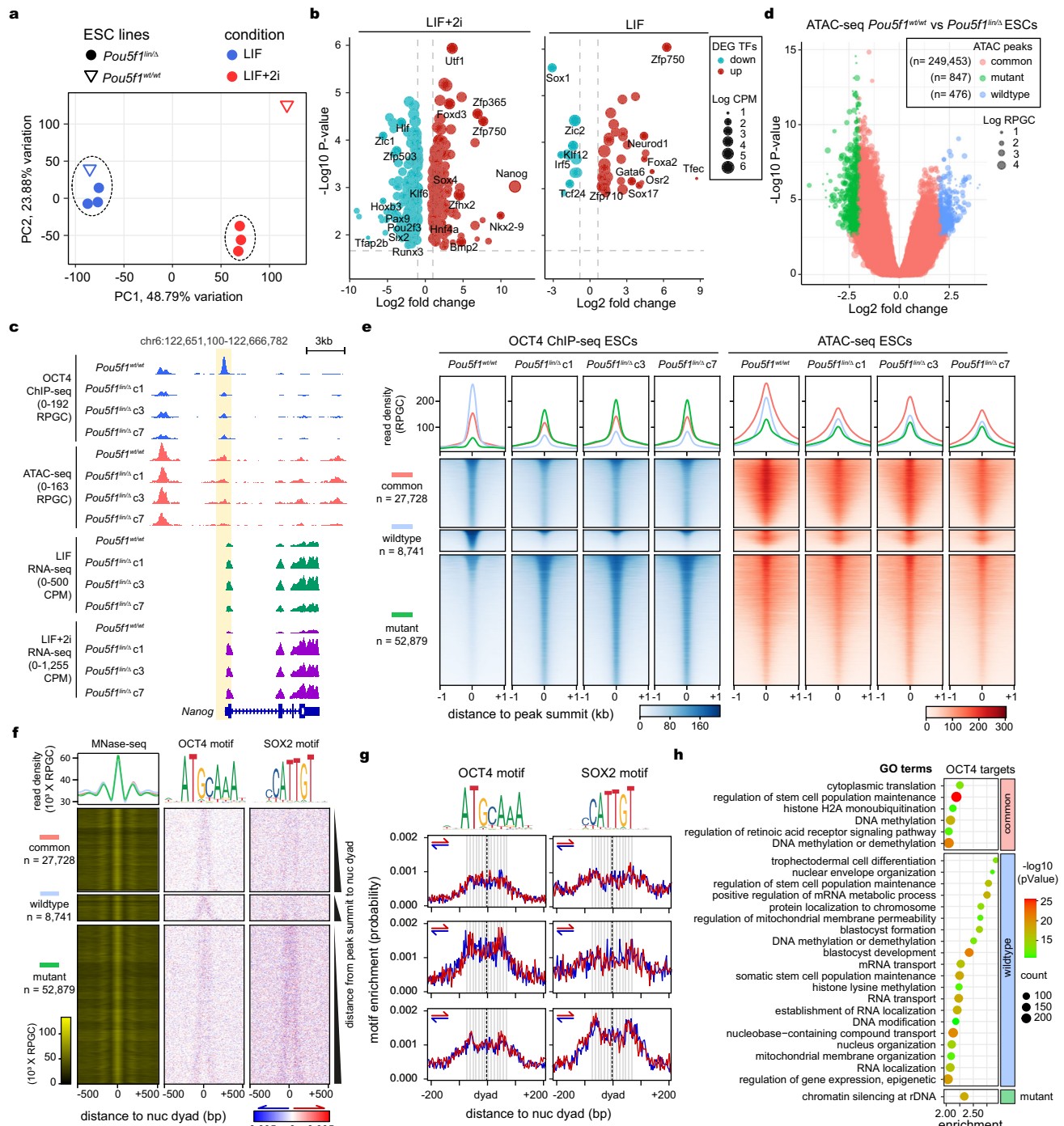

**Fig. 7 | Removing SLiPERs drives aberrant OCT4 binding in ESCs. a** Principal component analysis of RNA-seq showing distinct gene expression of *Pou5f1*[lin/Δ] ESC lines compared to *Pou5f1*[wt/wt] under LIF+2i and LIF conditions. **b** Volcano plots showing TF DEGs between *Pou5f1*[lin/Δ] and *Pou5f1*[wt/wt] ESCs under LIF+2i and LIF conditions. The significance threshold is represented by dotted lines denoting log2 (fold change) of 1.5 and FDR adjusted *P* value of 0.05 by unpaired *t*-test. **c** Genome browser screenshot showing OCT4 ChIP-seq, ATAC-seq and RNA-seq around the *Nanog* locus in *Pou5f1*[lin/Δ] and *Pou5f1*[wt/wt] ES cells. OCT4 binding is diminished in *Nanog* promoter is highlighted in yellow. **d** Volcano plot showing differential chromatin accessibility measured by ATAC-seq in *Pou5f1*[lin/Δ] and *Pou5f1*[wt/wt] ES lines. The significance threshold is represented by green and blue circles denoting; log2 (fold change) of 2 and FDR adjusted *P* value of 0.001 by unpaired *t*-test. The number of peaks (*n*) of each category are indicated. **e** ChIP-seq read density heatmaps (blue) of OCT4 WT and mutants in ES cells relative to ATAC-seq (red) spanning ±1 kb from

the centre of OCT4 peaks. The corresponding profile plots are shown above. The number of sites (*n*) of each group and the enrichment colour scale (RPGC) are indicated. **f** Read-density heatmaps of MNase-seq around nucleosome dyads (±500 bp) bound by OCT4 as in (**e**). The sites were rank ordered based on the distance between OCT4 peak summits and the closest nucleosome dyad. Density heatmaps of de novo motifs (logos on top) around nucleosome dyads. Motif density is scored on both DNA strands (red and blue) following the colour gradient scale shown at the bottom. The number (*n*) of nucleosomes are indicated. **g** Average profile plots of motif density scores on the two DNA strands (red and blue) around nucleosome dyads (±200 bp) targeted by OCT4 as in (**f**). Weighted frequency values were generated using kernel smoothing in 3 bp windows. DNA 10 bp twists around nucleosome are shown in grey-white stripes. **h** GO terms associated with genes targeted by OCT4 in the three binding site groups shown in (**e**).

pluripotency maintenance, distinguishing between the two processes. OCT4 SLiPERs are crucial for reprogramming but can be dispensable for supporting ESC self-renewal. Interestingly, OCT4 SLiPERs are required for ESCs to contribute normally to embryonic development beyond gastrulation. Our data support the notion that OCT4 adapts its IDRs not only to induce pluripotency but also to promote lineage commitment, suggesting a similar mechanism at play[16,19–21,26]. Interestingly, removing SLiPERs causes aberrant OCT4 binding in both reprogramming and ESCs' self-renewal. During reprogramming, removing SLiPERs causes OCT4 to target accessible sites mainly enriched for zinc-finger motifs. However, removing the same domain during ESCs self-renewal, causes OCT4 aberrant binding to SOX2 enriched motifs. Therefore, these domains contribute to the specific binding of OCT4 to the genome in a context-dependent manner despite being outside the OCT4-DBD. However, it's important to confirm the role of OCT4-SLiPERs in development. For example, rescue experiments using OCT4-SLiPER mutant fused to TNIP2, similar to what we observed in reprogramming, could provide further evidence. It will also be interesting to reveal if OCT4-SLiPERs are involved in transient reprogramming during embryonic development or adult stem cells[91,92].

Our results demonstrate that OCT4-SLiPERs influence genomic interactions by serving as recruitment interfaces. Notably, OCT4 colocalises with different proteins on chromatin during early reprogramming and ESC self-renewal, highlighting the distinct roles of OCT4 in these two processes. It is intriguing to find that UFD1L and TNIP2 are translocated from the cytoplasm to chromatin through their interaction with OCT4-SLiPERs during reprogramming. Such interactions enable OCT4 to exploit the unusual transactivation role of TNIP2 during reprogramming[79]. We also propose that OCT4 may exploit the function of UFD1L in relieving protein-induced chromatin stress by protein extraction during reprogramming[93]. This may also help somatic cells wipe out the somatic chromatin state before establishing pluripotency.

Recent studies have demonstrated the prevalence of IDRs, particularly among chromatin-associated proteins such as TFs[94–96]. Moreover, IDRs have been shown to contribute to TFs cell-type-specific binding[96,97]. Although intrinsically disordered, certain regions within IDRs can fold under specific conditions, such as during interactions with protein partners[60,98–100]. Interestingly, AlphaFold2 predicted that OCT4 has six quasi-ordered domains within IDRs, which overlap precisely with our OCT4-SLiPERs. We propose that these OCT4-IDR segments serve a cell-type-specific function by acting as interacting modules like MoRFs and SLiMs[96,99,101]. In conclusion, by incorporating function-specific modules within the otherwise generic IDRs, TFs can achieve both functional specificity and plasticity. It will be interesting to discover whether other TFs contain interchangeable IDR segments that can be used to change TF cell-type specificity in reprogramming.

## Methods
### Ethical compliance
hESCs MasterShef-7 line was established in Prof. Harry Moore's laboratory at the University of Sheffield. The use of hESCs was approved by the Steering Committee for the UK Stem Cell Bank and for the Use of Stem Cell Lines (ref: SCSC15-09). Human primary fibroblast cell lines were derived in RBiomedical, Edinburgh, UK, from skin samples from anonymous donors undergoing routine surgery at the Edinburgh Royal Infirmary, Little France, UK, under ethical approval 09/MRE00/91. All reprogramming experiments have been approved by the University of Edinburgh SBS ethics committee (asoufi-0001). All animal experiments for the iPSC generation from MEFs and chimera assays were approved by the University of Edinburgh Animal Welfare and Ethical Review Body, performed at the University of Edinburgh, and carried out according to regulations specified by the Home Office and Project License. Teratoma assays were performed in compliance with the joint ethics committee (IACUC) of the Hebrew University and Hadassah Medical Center and the National Ethics Committee (Israel Health Ministry) and NIH, which approved the study protocol for animal welfare. The Hebrew University is an AAALAC International-accredited institute. All animals were housed and treated in accordance with the veterinary guidelines and regulations of the University of Edinburgh and the Hebrew University.

### DNA constructs
The 119 OCT4 deletion mutants were synthesized as linear DNA fragments by Twist Bioscience (San Francisco, CA), each containing five amino acid deletion tiled with a two amino acid overlap (see Supplementary Data 1 for DNA sequence) and flanked by two adaptors for sub-cloning. The DNA fragments were amplified by PCR using primers containing EcoRI and NotI sites for cloning (see Supplementary Data 2 for sequences). The amplified PCR products were digested EcoRI and NotI and then ligated into a linearized pET28B vector (Novagen, Merck Millipore). The resulting plasmid library was verified by Sanger sequencing to confirm the presence of the appropriate deletion in each OCT4 mutant.

The lentivirus vectors used in iPSC reprograming studies were constructed by sub-cloning OCT4 variants into the FUW-tet-O plasmids. Each OCT4 deletion mutant was amplified from the corresponding pET28-OCT4 plasmid by PCR using set of primers containing EcoRI and XbaI (see Supplementary Data 2 for sequences). The amplified products were digested with EcoRI and XbaI and cloned into linearized FUW-tet-O-XbaI plasmid, which was modified from FUW-tet-O-hOct4 plasmid (Addgene plasmid; #20726)[102], by engineering an XbaI site to replace the EcoRI site (nucleotides 4139 to 4144) using a QuikChange II site-directed mutagenesis kit (Stratagene). For pluripotency rescue experiments, the OCT4 deletion mutants were PCR-amplified from the pET28-OCT4 deletion library and sub-cloned into pCAG-IRES-Puro plasmids[37] using the In-Fusion HD Cloning Plus kit according to the manufacturer's instructions (Takara Clonetech) and the appropriate primers (see Supplementary Data 2 for sequences).

The selected nine alanine stretch mutants of OCT4, in which a contiguous sequence of five alanine residues substituted for five naturally occurring residues, were generated by site-directed mutagenesis of pET28b-OCT4 plasmid using QuikChange II kit (Stratagene). The alanine stretch derivatives of OCT4 were subsequently sub-cloned into FUW-tet-O-XbaI and pCAG-IRES-Puro plasmids as describe above.

The truncated and linker substitution OCT4 mutants were all synthesized as DNA fragments and cloned into a Twist vector by Twist Bioscience (San Francisco, CA). The resulting OCT4 mutants were subsequently sub-cloned into pET28B, FUW-tet-O-XbaI and pCAG-IRES-Puro plasmids as describe above.

For knocking out gene candidates in mouse reprogramming, sgRNA sequences were obtained from a previously published sgRNA mouse library[103]. For cloning, two complementary oligonucleotides containing 19 bp gRNA sequence with "CACCG" appended to the 5′ end of the top strand and "AAAC" to the 5′ of the bottom strand as well as "C" at the 3′ end of the bottom strand (see Supplementary Data 3 for sequence). The complementary DNA Oligos were annealed by heating at 95 °C for 12 min followed by slow cooling at RT overnight. Annealed oligos were diluted to 7.1 pmol/μL and cloned into the vector pKLV2-U6gRNA5(BbsI)-sEF1aBFP-W (KPL474) into the compatible BbsI sites.

The human FUW-TetO-OCT4 (Addgene plasmid #20726), FUW-TetO-SOX2 (Addgene plasmid #20724), FUW-TetO-KLF4 (Addgene plasmid #20725) and FUW-TetO-CMYC (Addgene plasmid #20723) lentivirus vectors were generated by the Rudolf Jaenisch lab (Hockemeyer et al., 2008) and obtained from Addgene. The pWPT-rtTA2M2 vector was generated by ligating the rtTA2M2 gene, isolated from the pUHrT 62-1 vector (Urlinger et al., 2000), into the pWPT-GFP backbone vector after removing GFP (Kim et al., 2013). pSIN4-EF2-O2S and pSIN4-

CMV-K2M plasmids were a gift from James Thomson and deposited in Addgene (Addgene plasmid #21162 and #21164, respectively). The FUW-TetO-GATA3 expression vector was generated by cloning the GATA3 open reading frame into the pMINI vector (NEB) and then restricted with EcoRI or MfeI and inserted into the linearized FUW-TetO empty plasmid.

## Cell culture

MEF cells were cultured in MEF medium containing GMEM medium supplemented with 10% FCS, 1 mM sodium pyruvate (#11360-039, Gibco), 1 mM glutamine (#25030-024, Gibco), and 5.7 ml MEM 100X Non-Essential Amino Acids Solution (#11140050, Gibco), 50 μM 2-mercaptoethanol (#11528926; Gibco) at 37 °C and 5% $CO_2$. Mouse ESCs and mouse iPSCs were cultured in MEF medium supplemented with 1000 μ/ml human LIF at 37 °C and 5% $CO_2$. Under LIF-2i conditions, mESCs medium was further supplemented with 1 μM PD0325901 and 3 μM CHIR99021. Cas9 TNG MKOS MEFs (Tg MEF) were cultured in plates coated with 0.1% gelatin/PBS and MEF media supplemented with 5 ng/mL FGF2 and 1 μg/mL Heparin 37 °C and 5% $CO_2$. Irradiated MEFs (iMEF) were grown in plates coated with 0.1% gelatin in PBS and cultured in MEF media at 37 °C and 5% $CO_2$. Human Fibroblasts were cultured in HFibs media containing GMEM (#G5154; Sigma Aldrich) medium supplemented with 10% FCS (#10270-106, Gibco), 5.7 ml MEM 100X Non-Essential Amino Acids Solution (#11140050, Gibco), 1 mM Sodium Pyruvate (#11360-039; Gibco), 1 mM glutamine (#25030-024, Gibco) and 100 μM 2-mercaptoethanol (#11528926, Gibco) at 37 °C and 5% $CO_2$. HEK 293 were cultured with HFibs medium without 2-mercaptoethanol. Human Embryonic Stem Cells (hESCs) were cultured in Essential 8™ Medium (#A2858501, Gibco) in plates coated with 4 μg/mL of Human Recombinant Laminin 521/DPBS Ca++, Mg++ (#800110, BioLamina; #14040133, Gibco) at 37 °C in an atmosphere containing 5% $CO_2$.

## Lentivirus production

To produce lentiviral supernatant, $7.5 \times 10^5$ HEK 293T cells were seeded per 10-cm dish (Corning Inc., Corning NY) and cultured in GMEM medium supplemented with 10% FCS, 1 mM sodium pyruvate (#11360-039; Gibco), 1 mM glutamine (#25030-024; Gibco), and 5.5 ml Non-Essential Amino Acids at 37 °C and 5% $CO_2$. After 24 h and cells reaching ~40% confluency, a transfection cocktail was prepared by mixing 2.5 μg lentiviral expression plasmid, 1.7 μg psPAX2 packaging vector, 0.8 μg envelope vector, 30 μl Fugene6 (#E2691, Omega) and 570 μl OPTI-MEM media (#31985070; Invitrogen). After 15 min incubation at room temperature, the transfection cocktail was added on the culture medium in a dropwise fashion. The medium was replaced by fresh medium 16 h after transfection. The cells were further cultured for 55 h at 37 °C and 5% $CO_2$. The supernatant containing virus was cleared from dead cells by spinning at $600 \times g$ and filtered through a 0.45 μm syringe filter (#SLHV033RS; Millipore). Final volume of recovered viral supernatant was measured and polybrene was added to a final concentration of 4.5 μg/ml. The viral supernatant was aliquoted and stored at −80 °C. The viral titre of lentivirus encoding sgRNAs was determined by infecting MEFs and counting the number of BFP cells by flow cytometry.

To produce concentrated lentivirus for ChIP-seq and ChIP-SICAP experiments, lentivirus supernatant was prepared from $3 \times 10^6$ HEK 293T cells by scaling up the method above. The supernatants were then cleared by spinning at $600 \times g$ and filtered through a 0.45 μm syringe filter (#SLHV033RS; Millipore). Viral supernatant was transferred to thin-walled ultracentrifuge tubes (#344058; Beckman Coulter). Viral particles were subsequently pelleted by ultracentrifugation (25,000 rpm using a SW 32 TI rotor, Beckman Coulter) in a Beckman Optima XPN ultracentrifuge. The pelleted virus was resuspended in 100 μl plain GMEM medium and incubated at 4 °C for 16 h. The viral titre was determined by infecting MEFs or human fibroblasts using tet-

O-GFP lentivirus. The number of GFP-positive colonies was then calculated by flow cytometry. The average titre of the lentivirus used in this study was $5 \times 10^8$ infectious units per ml (IFU).

## Mouse iPSC reprogramming

Early passage MEFs ($5 \times 10^5$ cells) were seeded on a 10-cm dish (#430167; Corning Inc.) and cultured for 48 h in MEF medium at 37 °C and 5% $CO_2$. Twenty-four hours prior to viral transduction, $4 \times 10^4$ cells were seeded to gelatinized 6-well plates (#3516; Corning Inc.). The cells were then transduced by replacing the culture medium of each well with lentivirus supernatant mix containing 0.5 ml of each of O, S, K, M and rtTA viral supernatants. After 8 h, the lentivirus medium was removed and replaced with fresh MEF medium. After 16 h the infected cells were in induced to express OSKM (induction day 0) by replacing the culture medium with reprogramming medium containing LIF and 1 μg/ml doxycycline. The medium was renewed every two days until the induction day 6, whereupon it was renewed daily. At induction day 10, the doxycycline was removed from the reprogramming medium and emerging iPSC-like colonies were further cultured for 4 days.

For iPSC colony counting, cells were fixed and immuno-stained for Nanog and DAPI on day 14. Whole wells were imaged at a resolution of 4 μm/pixel using a CELIGO image cytometer (Nexcelom). DAPI-positive colonies were counted if covering an area larger than 8 adjacent pixels. Nanog-positive colonies were counted if at least 10% of the DAPI signal overlapped with the Nanog signal.

For generating clonal iPSC lines, individual iPSC colonies were picked on day 14 (see above) and transferred to a U-bottom 96-well plate containing 0.25% Trypsin-EDTA (#25200056; Gibco). After 5 min incubation at 37 °C, trypsinized cells were seeded to a gelatinized 24-well culture plate (#CLS3527; Corning). Undifferentiated iPSCs were subsequently transferred to larger culture dishes as they became confluent. At least 4 independent cell lines were generated from each condition and cryo-preserved using Mr. Frosty containers (#C1562; Sigma-Aldrich).

## Human iPSC reprogramming

Early passage (P5) human adult fibroblasts ($5 \times 10^5$ cells) were seeded on a 10-cm dish (#430167; Corning Inc.) and cultured for 3 days in HFibs medium at 37 °C and 5% $CO_2$. Twenty-four hours prior to viral transduction, $1 \times 10^5$ cells were seeded per well of a 6-well plates (#3516; Corning Inc.). On the day of transduction, the culture medium was changed to the OSKM viral supernatant mix containing 1,5 ml OCT4-2A-SOX2 and 1.5 ml KLF4-2A-cMYC viral supernatants. Viral supernatant was replaced with fresh HFibs medium at 8 h post transduction. After 4 days transduction, $5 \times 10^4$ infected HFibs were plated on iMEFs (A34181, Gibco), which were prepared a day earlier by seeding $3 \times 10^5$ cells per well on a 6-well plate. The reprogramming cells were cultured in hESC media, which was renewed daily, and incubated at 37 °C, 5% $CO_2$ and 3% $O_2$. Human iPSC colonies became visible at day 9-10.

To measure the reprogramming efficiency, colonies were fixed and immunostained for NANOG and DAPI on day 18. Human iPSC colonies were detected by imaging whole wells at a resolution of 4 μm/pixel using a CELIGO image cytometer. Nanog-positive colonies were counted if at least 10% of the DAPI signal overlapped with the Nanog signal.

To generate a human iPS lines, adult HFibs were reprogrammed as explained above but emerging hiPSC colonies were cultured for 21–23 days in hES medium. The colonies were collected by incubating with Accutase for 10 min at 37 °C and then physically separated from iMEFs using a sterile hypodermic needle. Collected colonies were seeded on laminin-coated plates and maintained in Essential 8 medium (Gibco, 15190617) for at least 8 passages before being used in subsequent experiments or cryo-preserved.

## Reprogramming human fibroblasts to hiTSCs

For infection, replication incompetent lentiviruses containing the OGKM reprogramming factors were packaged with a lentiviral packaging mix (7.5 µg psPAX2 and 2.5 µg pDGM.2) in 293T cells and collected 48, 60, and 72 h after transfection. The supernatants were filtered through a 0.45 µm filter, supplemented with 8 µg/ml of polybrene, and then used to infect human fibroblasts. Twelve hours following the fourth infection, medium was replaced with basic reprogramming medium (BRM) consisting of fresh DMEM containing 10%FBS, 1% L-glutamine and 1% penicillin-streptomycin. Six hours after medium replacement 2 µg/ml doxycycline (dox) was added to the medium. The reprogramming medium was changed every other day. After 14 days in BRM with dox, the medium was replaced with 50% BRM and 50% hTSC medium as described in ref. 104 supplemented with dox, followed by 7 days in 100% hTSC medium with dox. Then, dox was removed and colonies were allowed to stabilize for 7–10 days. Plates were then screened for primary hiTSC colonies, and then trypsinized for gene expression analysis.

For analysis of mRNA expression using qPCR, total RNA was isolated using the Macherey-Nagel kit (Ornat). 500–2000ng of total RNA was reverse transcribed using iScript cDNA Synthesis kit (Bio-Rad). Quantitative PCR analysis was performed in duplicates using 1/100 of the reverse transcription reaction in a StepOnePlus (Applied Biosystems) with SYBR green Fast qPCR Mix (Applied Biosystems). Specific primers were designed for the different genes. All quantitative real-time PCR experiments were normalized to the expression of GAPDH and presented as a mean ± standard deviation of two duplicate runs.

## Rescuing mouse ES cell self-renewal

ZHBTc4.1 ES cells were maintained in GMEM supplemented with 10% FCS, 1 mM glutamine/pyruvate, Non-essential Amino Acids, Penicillin/ Streptomycin and 0.05 mM 2-mercaptoethanol on porcine gelatine-coated 6-well plates. One million cells were seeded and transfected 3–4 h later with 2.5 µg plasmid DNA using Lipofectamine 3000 (#L3000008; ThermoScientific) in OptiMEM reduced serum medium (#31985062; ThermoScientific) following the protocol recommended by the manufacturer. After 24 h, cells were prepared for induction by harvesting and re-plating in two series of triplicate wells ($10^5$ cells per replicate, 3 for +dox, 3 for -dox). Induction was performed 3–4 h later by adding 1 µg/ml doxycycline and 2 µg/ml puromycin for selecting transfected cells. No doxycycline was added to the -dox control wells. The medium was then replaced every second day for a total of three changes. Representative pictures of growing colonies were obtained on the sixth day of culture. On the seventh day of culture, colonies were fixed and stained using an Alkaline Phosphatase (AP) Staining kit (#86R-1KT; Sigma-Aldrich) according to the manufacturer's instructions. AP-positive colonies were manually counted with the aid of a dissection microscope. Colony morphology, size and AP-positive red staining were used as scoring criteria, which were consistent between test and control plates, were as follows. In highly dense areas, touching colonies were counted as one to avoid over-estimation. The Rescue Index was calculated as the mean ratio of +dox counts/mean -dox counts. Error bars represent the standard deviation from the mean. All experiments were performed using cell batches of similar passage number for comparable results.

## Establishment of *Pou5f1* knock-in mouse ES cell lines

E14tg2α mES cells (early passage 4) were cultured in ES media on 0.1% gelatin coated 10-cm plates. The mES cells were cultured until forming rounded colonies and the media was changed 4–12 h prior to transfection. CRISPR-cas9 RNP complexes were prepared for four different reactions: (1) *Pou5f1* knock-in experiment containing *Pou5f1* gRNA and sDNA repair template, (2) *Pou5f1* knockout containing *Pou5f1* gRNA, (3) non-specific control containing scrambled gRNA, and (4) transfection control containing GFP plasmid. For each reaction, 1.2 µl of crRNA

(100 µM, IDT) was mixed with 1.2 µl of tracrRNA (100 µM, IDT) and annealed using the following PCR program: 95 °C with a normal ramp rate of 3–6 °C/s, 95 °C to 25 °C with a ramp rate of 0.1 °C/s, and 25 °C to 4 °C (hold) with a ramp rate of 0.5 °C/s. The Pou5f1 targeting crRNA sequence is shown below:

Mm.Cas9.Pou5f1.1.AX: 5′-ACGAGTGGAAAGCAACTCAG-3′

The Cas9 electroporation mix was prepared by combining the 2.4 µl of crRNA/tracrRNA mix with 1.1 µl of recombinant Cas9 protein (9 µg/µl), and 6.5 µl of single-stranded DNA (ssDNA, 120 µM, IDT) for the knock-in reaction. The mixtures were incubated at room temperature for 10 min and then placed on ice. The ssDNA sequence used as a template for HDR is shown below:

Pou5f1-lin29-42_homology_50bp: TCTCACCCCCACCAGGTGGGG GTGATGGGTCAGCAGGGCTGGAGCCGGGCGgCggaggCagcggaggCgga agcGGTGGGCCTGGAATCGGACCAGGCTCAGAGGTATTGGGGATCTC CCCATG

For each reaction, $7 \times 10^5$ freshly harvested mES cells were resuspended in 90 µl pre-warmed mES cell Nucleofector® mixture (20 µL Supplement per 90 µL Nucleofector® Solution) (#VPH1001, Lonza Biosciences). The cell suspensions were transferred to Amaxa-certified cuvettes, and electroporation was performed using the Nucleofector® II (Amaxa) program A-30/A-030 (Lonza Biosciences). Immediately after electroporation, cells were transferred to 15 ml tubes containing 5 ml of fresh mES media, centrifuged at 1200 rpm for 3 min, the medium was aspirated, and cells were resuspended in 2 ml of fresh media. Each electroporation reaction was transferred into one well of 6-well plates and incubated at 37 °C. Media was changed ~16 h post electroporation.

The transfected cells were harvested using TrypLE (Thermo-Scientific) when reaching ~90% confluency, ~48 h post electroporation. Half of the cells were harvested for genomic DNA (gDNA) extraction, while the other half was plated as single cells for establishing clonal mES lines. DNA extraction was performed using the DNeasy® Blood & Tissue Kit (Qiagen), following the manufacturer's protocol. DNA was eluted in 70 µl of elution buffer, and the concentrations were measured using a Nanodrop spectrophotometer (ThermoScientific). For assessing knock-in and knout efficiencies, a 519 bp DNA fragment containing the targeted *Pou5f1* region (a.a. 29-42), was amplified by PCR using the Phusion Plus DNA polymerase (#F631S, ThermoScientific), the extracted gDNA as a template, and the following primers:

Pou5f1-start-FWD: 5′-ATGGCTGGACACCTGGCTTC-3′,
Pou5f1-exon1-RVS: 5′-CTAGTCCACACTGCGTCGTGCT-3′

To evaluate the CRISPR knock-out and knock-in, the resulting PCR DNA product was sequenced and analysed by TIDE and TIDER assays[105].

To establish clonal ES-lines, the remaining transfected mES cells were resuspended in PBS containing 2% FCS (FACS buffer) to a final density of $1 \times 10^6$ cells/ml using BioRad TC-10 cell counter (Bio-Rad). The cells were sorted into 5×96-well plates (0.1% gelatin coated) using BD FACSA-ria II cell sorter (BD Biosciences) and cultured in mES media for ~15 days at 37 °C and 5% $CO_2$, changing the media every ~3 days. Around 30 colonies were harvested from each 96-well plate using TrypLE and the third of each colony culture was transferred into a one well of 24-well plate coated with 0.1% gelatin. The remaining two thirds of cell culture was used for gDNA extraction in 25 µl lysis buffer (25 mM KCl, 5 mM Tris pH 8, 1.25 mM MgCl2, 0.2% NP40, 0.2% Tween-20, 0.4 µg/µl Proteinase K). Next, the cell lysates were heated to 65 °C for 6 min and then to 98 °C for 5 min. Subsequently, the cell lysates were diluted by adding 75 µL AE elution buffer (Qiagen mini-prep). For PCR, 5 µl of the cell lysates (gDNA) was used in a 12.5 µl PCR reaction.

To screen for colonies containing the Pou5f1-lin29-42 knock-in mutation, two PCRs were performed for each colony using the Pou5f1-mutant-specific-FWD and Pou5f1-WT-specific-FWD primers, each paired with the Pou5f1-exon1-RVS primer. The gDNA extracted form wildtype mES cells and the pool transfected cells were used as controls. Colonies that generated a PCR product only when using the

Pou5f1-mutant-specific-FWD primer, were considered to contain mutations within Pou5f1 in both alleles. The primer sequences are listed below:

> Pou5f1-mutant-specific-FWD: 5′-GGAGGCAGCGGAGGCGGAAG-3′
> Pou5f1-WT-specific-FWD primers: 5′-GAACCTGGCTAAGCTTCCAA-3′
> Pou5f1-exon1-RVS: 5′-TAGGCCCAGTCCAACCTGAGGTC-3′

ES colonies with potential Pou5f1-lin29-42 knock-in mutation as assed by PCR were further expanded and the levels of OCT4 protein was measured by Western blot analysis. Only colonies that showed similar OCT4 protein levels to wildtype were taken forward for genotype analysis. Next, gDNA was extracted from the harvested cell pellets using the Qiagen DNeasy Blood & Tissue Kit (Qiagen), which was used to amply a 519 bp DNA fragment as explained above. The resulting DNA fragment from each mES colony was inserted into the pCR®-Blunt II-TOPO® vector using the Zero Blunt™ TOPO™ Cloning Kit following the manufacturer's protocol (ThermoScientific). Briefly, the PCR product (4 μl), Salt Solution (1 μl) and the TOPO vector (1 μl) were mixed and incubated for 10 min at room temperature, after which 2 μl from each reaction was added per vial of One Shot® chemically competent *Escherichia coli* cells and incubated for 30 min on ice. Afterwards, cells were heat-shocked at 42 °C for 30 s and allowed to recover by adding S.O.C. medium (250 μl) and incubating for 1 h at 37 °C with shaking. After recovery, 50 μl of transformed bacteria were plated on agar-coated 10 cm dishes containing kanamycin (1 μl/mL) and incubated at 37 °C overnight. The following day, individual colonies were picked and resuspended in nuclease-free water (15 μl). The cell suspension was used for colony PCR using M13 primers. The resulting PCR products were sent for Sanger Sequencing (Source BioSciences). The DNA sequences were analysed using SnapGene (Domatics) and each mES clone genotype was identified.

### Mouse chimaera

F1 female mice were superovulated (100 IU/ml PMSG, ProSpec, and 100 IU/ml HCG, Intervet, intraperitoneal injections 48 h apart) and crossed with wild-type stud male mice. Pregnant mice were culled at 2.5 dpc. by cervical dislocation, ovaries with oviducts were dissected and collected in pre-warmed M2 medium (Sigma-Aldrich). Oviducts were flushed using PBS and a 20-gauge needle attached to a 1 ml syringe and filled with PB1 medium. The blastomere-stage embryos (2.5 dpc) were collected and washed in PB1 medium, the proteinaceous zona pellucida was removed using acidic Tyrode's solution (Sigma-Aldrich). The zona pellucida–free embryos were washed with KSOM and transferred to a plate with depression wells (aggregation dish). A small clump (8–15 cells) of mES cells (donor GFP-positive) were added to each embryo using a 50 μm transfer pipette. Embryos were cultured in KSOM (Sigma-Aldrich) overnight at 37 °C in 5% CO$_2$. Blastocysts were selected and transferred into the oviduct of a 2.5 dpc pseudopregnant CD-1 female. Embryos were dissected in M2 medium and observed for chimeric ES cell contribution under an Olympus IX51 microscope, prior to fixation and immunostaining.

For some experiments, embryos were matured into blastocysts following 24 h of culture in KSOM (Sigma-Aldrich) in vitro, then fixed, processed for immunohistochemistry, and examined by confocal microscopy.

### Immunofluorescence and confocal microscopy of embryos

Blastocysts and pre-implantation embryos were fixed with 4% PFA in PBS for 15 min at room temperature, rinsed in PBS containing 3 mg/mL polyvinylpyrrolidone (PBS/PVP) and permeabilized with PBS/PVP containing 0.25% Triton X-100 for 30 min. Blocking was performed in embryo blocking buffer comprising PBS supplemented with 0.1% BSA, 0.01% Tween20 and 3% donkey serum for 2–3 h at 4 °C. They were inmunostained and rinsed three times for 15 min in blocking buffer. Blastocysts and pre-implantation embryos were transferred in small drops of blocking buffer onto poly-D-lysine-coated glass-bottom dishes under mineral oil and imaged using a Leica SP8 confocal microscope. Images were processed using Fiji.

Embryos were fixed with 4% formaldehyde/PBS/0.1% Triton X-100 (Sigma-Aldrich) for 20 (peri-implantation) or 30 (post-implantation) min and quenched with 50 mM ammonium chloride. Permeabilization was carried out for 10 min in PBS/0.1% Triton X-100. The embryos were incubated in primary antibody in 3% donkey serum/PBS/0.1% Triton X-100 (Nanog eBioscience, #14-5761-80; GFP Abcam, #ab13970; LamnB Sigma, #L9393) overnight and subjected to 3 washes in PBS/0.1% Triton X-100. Secondary antibodies were applied subsequently for 2 h to overnight, followed by 3 washes in PBS/0.1% Triton X-100. Embryos were then stained with DAPI (Biotium), mounted in PBS droplets covered with mineral oil in microscope rings, and imaged on a Leica SP8 confocal microscope. Following staining, chimaeric embryos were dehydrated in methanol series in PBS/0.1% Triton X-100, clarified in 50% methanol/50% BABB (benzyl alcohol:benzyl benzoate 1:2 ratio, Alfa Aesar and Sigma), transferred into 100% BABB into ibidi plates and imaged on a Leica SP8 confocal microscope.

### Immunostaining

Cells were fixed in phosphate-buffered saline (PBS) containing 4% paraformaldehyde for 10 min at room temperature. Fixed cells were permeabilized with 0.1% Triton X-100 for 10 min at room temperature and blocked with 4% Donkey (#G9663; Sigma-Aldrich) or Goat (#G9023; Sigma-Aldrich) serum in PBS for at least 60 min at room temperature. Blocked cells were incubated overnight in blocking buffer containing an appropriate concentration of primary antibody: anti-OCT4 1 μg/ml (#ab19857; Abcam), anti-SOX2 0.5 μg/ml (#AF2018; R&D systems), anti-KLF4 0.5 μg/ml (#AF3640, R&D systems), anti-MYC 1 μg/ml (#AF3696; R&D systems), anti-Nanog 0.5 μg/ml (#14-5761-80; eBioscience), or anti-NANOG (ab109250, Abcam). Antibody stained cells were washed 3 times with PBS containing 0.05% Tween-20. After washing, cells were incubated with an appropriate secondary antibody in PBS: donkey Anti-Rat IgG H&L 1 μg/ml (#A21208; Invitrogen), donkey anti-Rat IgG H&L 1 μg/ml (#A21209; Invitrogen), donkey Anti-Goat IgG H&L 1 μg/ml (#ab150129 Abcam), donkey Anti-Goat IgG H&L 1 μg/ml (#ab150130, Abcam), Goat Anti-Rabbit IgG H&L 1 μg/ml (#ab150077; Abcam), or Goat Anti-Rabbit IgG H&L 1 μg/ml (#ab150080; Abcam) for 2 h at room temperature. Nuclei were stained with 3 mg/ml 4,6-diamidino-2-phenylindole (DAPI) (#D3571; Invitrogen) for 10 min at room temperature. Fluorescence images were taken using the IRIS Digital Cell Imaging System (I10999; Logos Biosystems, Anyang, South Korea). Or whole wells were imaged at a resolution of 4 μm/pixel using a CELIGO image cytometer (Nexcelom, Lawrence, MA).

### RNA isolation and quantitative PCR analysis

RNA was harvested from cells using a RNeasy Mini Kit (#74106; Qiagen). The concentration of the harvested RNA was measured using a NanoDrop device. cDNA was synthesized from 500 ng RNA using High-Capacity cDNA Reverse Transcription kit (#4368814; Applied Biosystems). A 15-times diluted cDNA sample was amplified using a Roche LightCycler® 480 Instrument II and the Power SYBR Green PCR mix (#4367659; Applied Biosystems) under the following conditions: 50 °C for 4 min, 95 °C for 10 min, followed by 40 cycles of 95 °C for 15 s, 60 °C for 15 s, 72 °C for 45 s. Sequences of the oligonucleotide primers used for qPCR are given in Supplementary Data 4. GAPDH expression was used for normalization.

### Early reprogramming human fibroblasts (OSKM48hr)

HFibs were plated at a density of $5 \times 10^5$ cells in 10 cm dishes or $1.5 \times 10^6$ cells in 15 cm dishes. After 24 h, HFibs were infected with concentrated OSKM and rtTA2M2 lentiviruses at a multiplicity of infection (M.O.I) of 5. After 48 h, the culture medium was replaced with induction medium containing fresh HFib media with 1 μg/mL DOX and cultured for further 48 h at 37 °C and 5% CO$_2$.

## Chromatin preparation for ChIP-SICAP and ChIP-seq

Chromatin fragments were prepared from 15 cm dishes of hES, HFibs dox-induced for 48 h with OSKM and uninfected HFibs cells. Cells were washed 3X with PBS at RT and left in PBS/MgCl$_2$. Cross-linking was performed by adding first 120 µL of fresh DSG (0.25 M stock in DMSO) (#20593, ThermoScientific) to 15 mL of PBS/MgCl$_2$ followed by a 45-min incubation at RT with constant moving. Cells were washed 3X with PBS for a second cross-linking by adding 3 ml of formaldehyde fixative solution (50 mM HEPES-KOH pH 7.5, 100 mM NaCl, 1 mM EDTA, 0.5 mM EGTA, 11% formaldehyde (#28906, Pierce)) to 30 ml of PBS followed by a 10 min incubation at room temperature with constant moving. Crosslinking was quenched by adding 1.65 ml of a 2.5 M glycine solution and incubating at room temperature for 10 min. Cells were lifted using a scraper and pelleted by centrifugation at 1350 × $g$ for 5 min at 4 °C. The crosslinked cell pellet was washed 3 times with 10 ml of ice-cold PBS. Finally, the pellet was flash frozen in liquid nitrogen and stored at −80 °C for subsequent chromatin preparation.

Before nuclear extraction, the cell pellet was thawed on ice for 2−3 h and resuspended in 10 ml filtered, ice-cold lysis buffer (50 mM HEPES-KOH pH 7.5, 140 mM NaCl, 1 mM EDTA, 10% glycerol, 0.5% NP-40, 0.25% Triton X-100 and 1 tablet of Complete protease inhibitor cocktail (Roche)). The suspension was gently mixed on a rocker at 4 °C for 10 min. The cells were disrupted using a 7 ml glass-Dounce homogeniser (80 strokes) on ice. The nuclei were pelleted by centrifugation (1350 × $g$ for 5 min at 4 °C) and washed with 10 ml ice-cold wash buffer (10 mM Tis-HCl pH 8, 200 mM NaCl, 1 mM EDTA, 0.5 mM EGTA and 1 tablet of Complete protease inhibitor cocktail (Roche)) for 10 min at 4 °C. The nuclei were collected by centrifugation and resuspended in 4 ml sonication buffer (10 mM Tis-HCl pH 8, 100 mM NaCl, 1 mM EDTA, 0.5 mM EGTA, 0.1% Na-deoxycholate, 0.5% N-lauroylsarcosine and 1 tablet of Complete protease inhibitor cocktail (Roche)). The resuspended nuclei were split into four aliquots in pre-chilled 1 mL millitubes containing AFA Fibre (#520130; Covaris) and sonicated using a Covaris M220 Focused-ultrasonicator. Each millitube was sonicated for 10 min and kept on ice for 30 min per sonication cycle for a total of 9 cycles for hES and 14 cycles for HF and OSKM 48 h. Sonicated chromatin was transferred to Protein Lobind tubes (#0030108094; Eppendorf). 100 µL of 10% Triton X-100 was added to each tube to increase solubility. Samples were then centrifuged (20,000 × $g$ at 4 °C for 10 min) and the supernatant pooled into a fresh tube. A 50 µL aliquot of each pooled sample was analysed to check the fragment size distribution and quantify the DNA content of the resulting sonicated chromatin using a Nanodrop spectrophotometer. Another 50 µL aliquot was retained to be used as input for ChIP analysis. The sonicated chromatin and the input sample were snap frozen in liquid nitrogen and stored at −80 °C.

## ChIP-seq

For each ChIP, 35 µl Dyna Protein G magnetic beads (#10004D, ThermoScientific) were washed three times with 1 ml blocking solution (0.5% w/v BSA in PBS Tween-20). The beads were mixed with 10 µg anti-human OCT4 antibody (#ab19857; Abcam) and incubated on a rotating platform at 4 °C for at least 6 h. A 40 µg aliquot of double cross-linked and sonicated chromatin was mixed with the antibody saturated beads and incubated on a rotating platform at 4 °C overnight. The ChIP beads were transferred to a fresh tube, washed with 4X with 1 mL RIPA buffer (50 mM HEPES-KOH pH 7.6, 500 mM LiCl, 1 mM EDTA, 1% NP-40, 0.7% Na-deoxycholate). The ChIP beads were washed once with TE buffer (10 mM Tris-HCl pH 8, 1 mM EDTA) containing 50 mM NaCl. The bound-chromatin was eluted by incubating the ChIP beads in 210 µl elution buffer (50 mMTris-HCl pH 8, 10 mM EDTA, 1% SDS) at 65 °C for 30 min. The beads were pelleted by centrifugation at 16,000 × $g$ for 1 min. 200 µl supernatant containing the soluble chromatin was then transferred to a fresh tube. A 50 µl aliquot of sonicated input DNA was thawed and diluted with 150 µl elution buffer. Soluble chromatin and input DNA samples were reverse crosslinked at 65 °C for 16 h. 200 µl TE was added to the soluble chromatin and input DNA to reduce the concentration of SDS. RNA in the samples were digested using 0.2 mg/ml RNAse A followed by incubation at 37 °C for 2 h. Protein was then digested using 0.2 mg/ml Proteinase K at 55 °C for 2 h. The ChIP DNA was subsequently purified by phenol-chloroform extraction followed by ethanol precipitation. The ChIP DNA pellet was resuspended in 21 µl TE buffer. ChIP DNA concentration was measured by Qubit Fluorometric Quantitation (#Q32854; ThermoScientific).

## ChIP-seq library generation

ChIP-DNA from three ChIP replicates were pooled together. The amount of DNA used in the library preparation ranged between 20 and 50 ng. Input DNA (20 ng) was generated by pooling equal amounts of input DNA from each condition. The DNA libraries were prepared using NEBNext Ultra II DNA Library Prep Kit (#E7645S; New England Biolabs) according to manufacturer's instructions. Each library was uniquely barcoded by using NEBNext Multiplex Oligos for Illumina (Dual Index Primers Set 1) (#E7600S; New England Biolabs) as indicated. Size selection (-200 bp) of the DNA library was carried out using SpeedBeads, which comprise magnetic carboxylate modified particles (GE65152105050250; ThermoScientific). The ChIP-seq DNA library pool was generated by mixing an equal volume of each library. The quality of the DNA library was assessed using Agilent HS DNA Screen Tape and Reagents (#5067-5585; Agilent). The concentration of the ChIP-seq DNA library pool was determined to be 23.19 nM by Qubit Fluorometric Quantitation (#Q33230, ThermoScientific). The ChIP-seq DNA library was submitted for sequencing using a NovaSeq S1 50PE (1 lane) platform (Illumina). Sequencing was performed by Edinburgh Genomics (Kings Buildings, The University of Edinburgh).

## ChIP-SICAP

Method was adapted from ref. 46. For each ChIP-SICAP, 70 µL of Protein G Dynabeads (#10004D, Invitrogen™) were washed 3X with PBS + 0.02% Tween. The beads were mixed with 20 µg anti-human OCT4 antibody (#ab19857; Abcam) or Rabbit-IgG isotype control for hES Control (#ab37415, Abcam) and incubated on a rotating platform at 4 °C for at least 6 h. A 120 µg aliquot of double cross-linked and sonicated chromatin was mixed with the antibody saturated beads and incubated on a rotating platform at 4 °C overnight. Beads were washed 1X with 10 mM Tris-HCl, 1X with 200 µL 1xTdT buffer (#EP0161, Thermo Scientific™) and resuspended in 92 µL of 1x TdT buffer. Biotynilation was performed by adding 4 µL ddUTP-Biotin (Jenabioscience, NU-1619-BIOX-S) and 4 µL of TdT enzyme (#EP0161, Thermo Scientific™) followed by an incubation of 30 min at 37 °C. Beads were washed 6X with ice-cold IP buffer (50 mM Tris-HCl pH 8, 5 mM EDTA, 1% Triton, 0.5% NP40, 150 mM NaCl). The bound-chromatin was eluted by incubating the ChIP beads in 100 µl of elution buffer (200 mM of DTT and 7.5% of SDS) and incubated at 37 °C for 30 min. Beads were pelleted by centrifugation at 16,000 × $g$ for 1 min. 100 µl supernatant containing the soluble chromatin was transferred to a fresh tube. For each replicate two ChIP were mixed at this stage, meaning each replicate consist of 240 µg of initial chromatin. Volume was adjusted to 1.4 mL with IP buffer and transferred to 50 µl of Streptavidin beads (#S1420S, NEB) (previously washed 3X with IP buffer) and incubated on a rotating platform 1 h at room temperature. Beads were then washed 3X with 1% SDS, 1X with 2 M NaCl, 2X with 20% isopropanol and 4X with 50% acetonitrile. FASP was done as described[106]. Briefly, beads were resuspended in 100 µL of 0.1% RapiGest (#186001861, Waters) and 25 mM DTT. Reverse-crosslinking and elution was performed by boiling the beads for 10 min at 95 °C. The beads were pelleted by centrifugation at 16,000 × $g$ for 1 min. and supernatant was transferred to a fresh. Urea in powder was added directly to the sample to a final concentration of 8 M and shaked at RT until dissolved. The solution was transferred to a Vivacon 500 spin column (30 K catridge,

#VN01H22, Sartorius) and centrifuged at RT for 15 min at 13,800 × $g$. Samples were alkylated in the membrane in 55 mM iodoacetamide (IAA) prepared in 8 M urea (dissolved in 0.1 M Tris-HCl, pH 8.2) for a 20 min at RT protected from light. IAA was washed from the membrane by centrifuging the spin column at 13,800 × $g$ for 15 min. Membrane was washed with 100 µl of 8 M urea and and 100 µL of 0.05 M of ammonium bicarbonate (ABC) buffer in ultrapure water, centrifuging for 15 min at 13,800 × $g$ between washes. Collection tubes were removed and the spin column was placed in a new Lo-bind tube. Protein digestion was performed by adding 1 µg Trypsin (#V5280, Promega) and 50 ng Lys-C (#125-05061, Wako) (in 100 µl of 5 mM ABC buffer) to the membrane column for further overnight incubation at 37 °C. Peptides were recovered by centrifuging the spin columns for 15 min at 13,800 × $g$. Supernatant with digested peptides was transferred to a fresh tube. Spin column was washed 1X with 100 µL ABC buffer and centrifuged for 15 min at 13,800 × $g$. Supernatant was collected and pooled with the first containing the digested peptides. Digested proteins solution were stored at −20 °C and until StageTip cleaning and mass spectrometry analysis.

## SILAC

HFibs and hESCs for SILAC were cultured using the SILAC Protein Quantitation Kit (LysC) – DMEM:F12 (# A33970, Thermo Fisher). HFibs were cultured in SILAC "heavy medium" consisting in DMEM:F12 for SILAC supplemented with 10% dialyzed Fetal Bovine Serum (FBS) for SILAC, 1 mM Sodium Pyruvate, 1 mM glutamine, 1X Non-Essential Amino Acids, 0.1 mM, 2-mercaptoethanol, 0.46 mM $^{13}C_6$ L-Lysine-2HCl (heavy) and 0.47 mM L-Arginine-HCl (light). hESCs were cultured in Laminin 521-coated plates with iMEF-conditioned SILAC "Light medium" consisting in DMEM:F12 for SILAC supplemented with 20% KnockOut Serum (#10828028, Gibco), 1 mM glutamine, 1X Non-Essential Amino Acids, 100 µM 2-mercaptoethanol, 0.46 mM L-Lysine-2HCl (light), 0.47 mM L-Arginine-HCl (light) and 10 ng/mL bFGF (#100-18B, Peprotech). Both "heavy and light" media were filtered (0.22 µm) before use. hESCs and HFibs were cultured in SILAC media for at least 5 passages in 10 cm dishes. Cells were scraped and pellets were collected by centrifugation for 5 min at 500 × $g$ and washed 2X with PBS. Cells were lysed on ice using 200 µL RIPA lysis buffer (#89901; Thermo Scientific) supplemented with Complete protease inhibitor cocktail (#04693124001; Roche) and sonicated on high with Bioruptor (6 × 10 s intervals). Suspension was incubated at 4 °C for 15 min. Cells lysates were centrifuged for 15 min at 13,800 × $g$ and 4 °C. Supernatant with protein extract was collected in a fresh tube. The protein concentrations of the lysates were quantified by Pierce BCA Protein Assay Kit (#23225, Thermo Scientific). 5 µg of protein were resolved by SDS-polyacrylamide gel electrophoresis and stained with GelCode Blue Safe Protein Stain. For each replicate, equal amount of HFibs and hESCs protein was mixed in a new tube for further in gel-digestion and mass spectrometry analysis.

## In gel digestion

For in-gel digestion, proteins were separated and concentrated on gel (NuPAGE Novex 4–12% Bis-Tris gel, Life Technologies), in NuPAGE buffer (MES) and visualised using InstantBlueTM stain (Sigma-Aldrich). The stained gel bands were excised into cubes ca. 1 mm and de-stained with 50 mM ammonium bicarbonate (Sigma Aldrich, UK) and 100% (v/v) acetonitrile (Sigma-Aldrich). Proteins were digested with trypsin, as described in Shevchenko et al.[107]. Briefly, proteins were reduced in 10 mM dithiothreitol (Sigma-Aldrich) for 30 min at 37 °C and alkylated in 55 mM iodoacetamide (Sigma-Aldrich) for 20 min at RT protected from light. Proteins were digested for 16 h at 37 °C with 12.5 ng µL$^{-1}$ trypsin (Pierce) in digestion buffer (1030 µL of dH2O, 300 µL of 50 mM ABC and 150 µL of ACN) making sure to cover all the gel pieces. Next day, digestion was stopped by acidifying the sample to pH < 2.5 with 10% Trifluoroacetic adic (TFA) and digested

samples were diluted with equal volume of 0.1% TFA. Supernatant liquid was recovered from the gel pieces and digested peptides were cleaned using StageTips.

## StageTip cleaning

All samples digested either by FASP (ChIP-SICAP) or in gel-digestion (SILAC) were cleaned using StageTips as described by Rappsilber et al.[108]. Briefly, peptides samples were passed and concentrated into manually cut Empore Disk C18 Octadecyl (C18)-bonded silica inserted into a pipette tip. Tips were left at −20 °C for further analysis by LC-MS.

## LC-MS and protein identification

Peptides were eluted in 40 µL of 80% acetonitrile in 0.1% TFA and concentrated down to 1 µL by vacuum centrifugation (Concentrator 5301, Eppendorf). LC-MS/MS was performed by diluting samples into 5 µL with 0.1% TFA. LC-MS-analyses were performed on an Orbitrap Fusion Lumos Tribrid and on a Q Exactive mass spectrometers (both from ThermoScientific) both coupled on-line, to Ultimate 3000 RSLCnano Systems (Dionex, ThermoScientific). For both approaches, peptides were separated on a 50 cm EASY-Spray column (ThermoScientific) assembled on an EASY-Spray source (ThermoScientific) and operated at a constant temperature of 50 °C. Mobile phase A consisted of 0.1% formic acid in water while mobile phase B consisted of 80% acetonitrile and 0.1% formic acid. Peptides were loaded onto the column at a flow rate of 0.3 µL min$^{-1}$ and eluted at a flow rate of 0.25 µL min$^{-1}$ according to the following gradient: 2 to 40% buffer B in 150 min, then to 95% in 11 min. For Orbitrap Lumos, survey scans were performed at 120,000 resolution (scan range 350–1500 $m/z$) with an ion target of 4.0e5. MS2 was performed in the Ion trap at rapid scan mode with ion target of 2.0$^e$4 and higher-energy collisional dissociation (HCD) fragmentation with normalized collision energy of 27[109]. The isolation window in the quadrupole was set at 1.4 Thomson. Only ions with charge between 2 and 7 were selected for MS2. For Q Exactive, MS1 spectra were recorded at 70,000 resolution and the top 10 most abundant peaks with charge ≥ 2 and isolation window of 2.0 Thomson were selected and also fragmented by HCD fragmentation of 27. The maximum ion injection time for the MS and MS2 scans was set to 20 and 60 ms respectively and the AGC target was set to 1 E6 for the MS scan and to 5 E4 for the MS2 scan. Dynamic exclusion was set to 6 s. Raw data files and peptide search was performed in The MaxQuant software platform[110] version 1.6.1.0 (released in April 2018). Search was conducted against the complete/reference proteome of Homo sapiens (released in November, 2017), using the Andromeda search engine[111]. The first search peptide tolerance was set to 20 ppm while the main search peptide tolerance was set to 4.5 pm. Isotope mass tolerance was 2 ppm and maximum charge to 7. Maximum of two missed cleavages were allowed. Carbamidomethylation of cysteine was set as fixed modification. Oxidation of methionine and acetylation of the N-terminal were set as variable modifications. Label-free quantitation analysis was performed by employing the MaxLFQ algorithm as described by Cox et al.[112]. SILAC quantification analysis was performed by employing the MaxQuant as described by Cox et al.[113]. For peptide and protein identifications, FDR was set to 1%. All MS were carried out by the Proteomics service (Wellcome Discovery Research Platform for Hidden Cell Biology) at the University of Edinburgh.

## Western blot analysis

Whole cell extracts were prepared by lysing cells using RIPA buffer (25 mM Tris HCl pH 7.5, 150 mM NaCl, 1% Na-deoxycholate, 1% NP-40, 0.1% SDS) supplemented with Complete protease inhibitor cocktail (#04693124001; Roche) and Halt phosphatase inhibitor cocktail (#78420; ThermoScientific). The protein concentrations of the lysates were quantified by BCA assay using a standard protocol. Proteins resolved by SDS-polyacrylamide gel electrophoresis were electro-blotted onto a PVDF membrane. The primary antibody incubations

with anti-human Oct4 antibody (0.45 mg/ml #ab19857; Abcam) and anti-human GAPDH (1:5000 dilution of GTX627408; GeneTex Inc, Irvine, CA) were performed at 4 °C for 16 h. The secondary antibody incubations with goat anti-rabbit IgG-HRP (1:5000 dilution of #sc-2004; Santa Cruz Biotechnology) and goat anti-mouse IgG-HRP (1:5000 of #sc-2005, Santa Cruz Biotechnology) were performed for 1 h at room temperature. Blots were visualized by using SuperSignal™ West Pico Chemiluminescent Substrate (#34080; Thermo Scientific) using a Mi5 Processor (MI-5 ADC, PGPR 14) imaging system.

## Sub-cellular fractionation

Subcellular fractions was performed as reported previously[114]. Briefly around one million cell pellets were lysed and resuspended with 5 volumes of ice-cold E1 Buffer (50 mM HEPES-KOH pH 7.5, 140 mM NaCl, 1 mM EDTA, 10% glycerol, 0.5% NP-40, 0.25% Triton X-100, 1 mM DTT, 1X cOmplete Ultra Protease Inhibitor) and centrifuged at $1100 \times g$ at 4 °C for 2 min. Supernatant was collected as the cytoplasm fraction. Remaining pellet was incubated in ice for 10 min with 5 volumes of E1. Nuclear pellet was recovered by centrifugation ($1100 \times g$ at 4 °C for 2 min) and gently resuspended in 2 volumes of ice-cold E2 buffer (10 mM Tris-HCl pH 8.0, 200 mM NaCl, 1 mM EDTA pH 8.0, 0.5 mM EGTA, 1X cOmplete Ultra Protease Inhibitor protease inhibitor cocktail). Nuclear extract was recovered by centrifugation ($1100 \times g$ at 4 °C for 2 min) and collection of supernatants. Remaining pellet was incubated in ice for 10 min with 2 volumes of E2 and centrifuged at $1100 \times g$ at 4 °C for 2 min. Chromatin fraction was prepared by resuspension of pellet in 2 volumes of ice-cold E3 buffer (500 mM Tris-HCl pH 6.8, 500 mM NaCl, 1 X cOmplete Ultra Protease Inhibitor protease inhibitor cocktail) for further sonication were for 5 min, 30 s ON/30 s OFF on maximum power (Bioruptor). All fractions (cytoplasm, nuclear and chromatin) were centrifuged for one last time at $16,000 \times g$ at 4 °C for 10 min. The protein concentrations of the lysates were quantified by Pierce™ BCA Protein Assay Kit (#23225, Thermo Scientific). Proteins resolved by SDS-polyacrylamide gel electrophoresis were electro-blotted onto a PVDF membrane. The primary antibody incubations with anti-human OCT4 antibody (0.45 mg/ml #ab19857; Abcam), anti-human GAPDH (1:5000 dilution of GTX627408; GeneTex Inc), anti-human TNIP2 (1:1000; #15459-1-AP, ProteinTech), anti-human ETV4 (1:1000; #PA5-79223, ThermoScientific), anti-human XPO6 (1 µg/ml, #ab72333, Abcam), anti-human UFD1L (1:1000; #10615-1-AP, ProteinTech), anti-human histone H3 (1:10,000; #ab24834, Abcam) and anti-human lamin (0.5 µg/mL; #ab16048, Abcam) were performed at 4 °C for 16 h. The secondary antibody incubations with goat anti-rabbit IgG-HRP (1:5000; #sc-2004, Santa Cruz Biotechnology) and goat anti-mouse IgG-HRP (1:5000; #sc-2005 Santa Cruz Biotechnology) were performed for 1 h at room temperature. Blots were visualized by using SuperSignal™ West Pico Chemiluminescent Substrate (# 34080; ThermoScientific) using an X-ray film developer (Amersham Hyperfilm #28906836; GE healthcare).

## CRISPR-cas9 knock-out in mESC and clonogenicity assay

Cas9-expressing TNG ESCs were seeded at a density of $5 \times 10^4$ cells per well in a 12-well gelatine-coated plate and were transduced the following day at M.O.I of 2 with lentiviral supernatant containing 8 µg/ml poly-brene (Millipore) for 6 h, after which infection medium was replaced with fresh ESC medium. Transduced ESCs were harvested 2 days later and BFP +ve cells were FACS-sorted (FACSAriaII, BD Biosciences) and plated at a clonal density (700 cells per well) in a 6-well gelatine-coated plate. Medium was replenished every other day. Whole-well imaging and quantification of Nanog-GFP+ colony numbers were performed with the Celigo S Cell Cytometer (Nexcelom) after 10 days in culture.

## CRISPR-cas9 knock-out in MEF and proliferation assay

CRISPR-Cas9-expressing MEFs were seeded at a density of $10^5$ cells per well in 6-well plate and were transduced the following day at M.O.I of 5 with a lentiviral supernatant containing 8 µg/ml polybrene (Millipore) for 4 h. The medium was then changed to fresh medium and cells were cultured for 2 days. Infection efficiency was confirmed to be above 90% in all samples by counting BFP +ve cells using FACS (FACSAriaII, BD Biosciences). MEFs were then harvested and stained with 2.5 µM of CellTrace™ Far Red dye (ThermoScientific) in PBS for 20 min at 37 °C and thoroughly washed with cold MEF medium. Cells were then plated at a density of $2 \times 10^4$ cells per well in 12-well plate and maintained in culture for 4 days. CellTrace™ Far Red-labelled cells at day 0 and day 4 were harvested and washed with FACS buffer (2% FCS in PBS) prior to acquisition with LSRFortessa (BD Biosciences) cytometer. Dead cells were excluded using SYTOX™ Green Nucleic Acid Stain (100 nM, ThermoScientific). Data were analysed using Flowjo v10 proliferation platform, the undivided mean was fixed using samples stained at day 0 and the division index (average number of divisions for all transduced in the original starting population) calculated on day 4 was extracted.

## Knock-down using siRNA

Human Fibroblasts (hFibs) $5 \times 10^5$ cells were seeded on a 10 cm dish and cultured in hFib media containing GMEM (#G5154, Sigma-Aldrich) supplemented with 10% FCS (#10270-106, Gibco), 1 mM Sodium Pyruvate (#11360-039, Gibco), 1 mM glutamine (#25030-024, Gibco), 0.05 mM Beta-mercaptoethanol (#31350010, Life technologies) and no-antibiotics at 37 °C and 5% $CO_2$ overnight. The next day, when cells reached ~40% confluency, hFibs were transfected (first round) by adding 2 ml of siRNA mixture in a drop-wise fashion to fresh hFibs media (no antibiotics). The siRNA mixture was prepared by mixing 1 mL Opti-MEM (#31985062, ThermoScientific) containing 15 µL of 20 µM siRNA stock with 1 mL Opti-MEM containing 30 µL Lipofectamin RNAiMAX transfection reagent (#13778075, ThermoScientific) and incubating for 15 min at room temperature. Transfection media was replaced with fresh hFibs media after ~6 h and cells were incubated at 37 °C and 5% $CO_2$ for 72 h. Then, $5 \times 10^5$ of siRNA-transfected cells were seeded on a 10 cm dish and cultured in hFib media at 37 °C and 5% $CO_2$ overnight. The cells were then transfected for a second round with siRNA as described above. A SMARTpool of siRNA targeting *TNIP2* and *STAT3* were ordered from Horizon/Dharmacon using L-014328-01-0005 and L-003544-00-0005, respectively.

## Luciferase assay

The pGL3 constructs containing the Fgf4 and Utf1 enhancers were obtained from the Niwa lab and were sequence-verified by Sanger sequencing. ZHBT4.C mES cells rescued with OCT4 wildtype or mutant (see the Rescuing mouse ES cell self-renewal section above) were co-transfected with a pGL3 construct and a Renilla expressing plasmid at a 10:1 w/w ratio using the Lipofectamine-3000 transfection reagent (ThermoScientific). For each transfection, 30,000 cells were seeded per well of a 24-well plate coated with 0.1% gelatin and cultured overnight. 48 h after transfection, luminescence was measured using the Dual-Luciferase Reporter Assay System (#E1910, Promega) according to the manufacturer's instructions, on a Promega Glomax Multi Detection System.

## Recombinant protein expression and purification

Recombinant proteins were expressed in *E. Coli* by transforming Rosetta™ 2 (pLysS) host strains (Novagen, Merck Millipore) with each of the recombinant pET28b constructs encoding wild-type hOCT4 or a mutant version of the protein fused to an N-terminal 6 x histidine tag. Cells were cultured at 37 °C in 5 mL of LB medium supplemented with chloramphenicol and kanamycin to an optical density at 600 nm of ~0.4. Heterologous gene expression was then induced by addition of IPTG to a final concentration of 0.5 mM and growth was continued at 37 °C for a further 5 h. The harvested cell pellets from a 5 mL culture were resuspended in PBS containing cOmplete™ EDTA-free Protease Inhibitor Cocktail (Roche). The cells were then disrupted by sonication

using a Bioruptor sonicator (Diagenode) on high power setting (10 × 30 s bursts with 30 s intervals). Sonication was performed in a cold room (4 °C) using chilled water in the sonication bath. The cell lysate was clarified by centrifugation at 5000 × g for 20 min at 4 °C and the pellet (insoluble fraction) collected. Subsequent purification steps were carried out at room temperature. The insoluble fraction was resuspended in 600 μl of denaturing buffer (DB) comprising; 8 M urea, 0.1 M NaH2PO4 (pH 8.0), 0.3 M NaCl, 10 mM imidazole and then applied to a HisSpinTrap column (GE Healthcare) pre-equilibrated in the same buffer. Sample application and washing was carried out according to the manufacturer's instructions. After washing the column with DB, bound proteins were eluted in 200 μl elution buffer (EB) (same as DB but containing 500 mM imidazole).

The purified denatured proteins were refolded by buffer exchange to remove EB and replace with refolding (RF) buffer (50 mM MES (pH 5.5), 240 mM NaCl, 10 mM KCl, 2 mM MgCl2, 2 mM CaCl2, 0.8 M urea, 30% (v/v) glycerol, 0.1% NP40 substitute, 0.05% (v/v) Triton X100, 2 mM EDTA, 5 mM DTT). Buffer exchange was performed using a PD-SpinTrap G25 column (GE Healthcare) equilibrated in RF buffer according to the manufacturer's instructions. A 180 μl aliquot of purified protein from the HisTrap column was applied to the PD-SpinTrap G25 column. The purified protein was aliquoted and snap frozen using liquid nitrogen before storage at −80 °C. Protein purity was assessed by sodium dodecyl sulfate polyacrylamide gel electrophoresis (SDS-PAGE) using the Bolt system with 4–12% gradient Bis-Tris gels (ThermoScientific), followed by protein staining using GelCode™ to visualise protein bands (ThermoScientific). Protein quantification was estimated by densitometry measurements of protein bands in digital images of SDS-PAGE gels stained with GelCode using Multi Gauge image software ver2.0 (FujiFilm), which were then plotted in a standard curve. Proteins with similar sizes and known concentrations were used to generate the standard curve. OCT4 wildtype was run with all OCT4 mutants in the same SDS-PAGE to reduce errors in protein quantifications.

## Nucleosome reconstitution
Nucleosome assembly was performed using the Cy5-labelled *LIN28B* DNA and recombinant human histones refolded to H2A/H2B dimers and H3/H4 tetramers as described previously[43]. Briefly, a 500 μl mixture containing 6 μg H2A/H2B, 6 μg H3/H4 and 10 μg Cy5-labelled *LIN28B* DNA in 4 M urea, 2 M NaCl and 1 mg/mL BSA was prepared. The mixture was dialyzed against 1 L denaturing buffer containing 5 M urea, 10 mM Tris HCl pH 8.0, 1 mM EDTA, 2 M NaCl, and 10 mM 2-mercaptoethanol at 4 °C overnight. The salt level in the sample was then gradually reduced by successive dialysis steps at 4 °C against 1 L of the same denaturing buffer containing 1.5 M NaCl for 2 h, 1.0 M NaCl for 2 h, 0.8 M NaCl for 2 h and then 0.6 M NaCl overnight. Next, the sample was dialyzed against 1 L of non-denaturing buffer containing 10 mM Tris HCl pH 8.0, 1 mM EDTA, 0.6 M NaCl, 1 mM 2-mercaptoethanol at 4 °C for 6 h. The sample was then dialyzed against the non-denaturing buffer containing 0.1 M NaCl at 4 °C overnight. The nucleosome preparation was centrifuged (15,000 × g at 4 °C for 10 min) and the pellet discarded. Finally, the nucleosomes were heat-shifted by incubation at 37 °C for 6 h. An aliquot of the nucleosome preparation was analysed on native 5% polyacrylamide gel alongside free *LIN28B* DNA to assess its quality and quantity by Ethidium Bromide staining and Cy5 fluorescence.

## Electrophoretic mobility shift assays
The binding to Cy5 end-labelled *LIN28B* DNA or oligonucleotide duplexes were analysed in native 4% or 5% polyacrylamide gels (dimensions: 0.15 × 18 × 18 cm), respectively, which were prepared in 0.5 X TBE (45 mM Tris-borate, 1 mM EDTA). Gels were stored overnight at 4 °C in 100% humidity before pre-running at 90 V (-10 V/cm) for 1 h. For affinity analysis, a 40 μl mixture typically containing 1 nM Cy5-labelled DNA and 0 to 10 nM purified wildtype or mutant OCT4 were prepared in 1 X binding buffer (10 mM Tris HCl pH 7.5, 1 mM MgCl2, 10 μM ZnCl2, 10 mM KCl, 1 mM DTT, 5% (v/v) glycerol, 0.5 mg/mL BSA). The mixtures were incubated at 20 °C ± 1 °C in the dark for 60 min using protein LoBind tubes (Eppendorf UK Ltd). A 30 μl aliquot of each sample was then loaded onto acrylamide gels and electrophoresis was performed at 90 V for 4 h. The gels were imaged by detecting Cy5 fluorescence using a Fujifilm Life Science FLA-5100 instrument (Fuji-Film). The resulting images were visualized, and the bands quantified using Multi Gauge image software ver2.0 (FujiFilm). For competition assays, the experiments were conducted in the presence or absence of a 40-fold excess of unlabelled competitor DNA (either specific DNA containing the cognate OCT4 binding site from the *FGF4* promoter or non-specific DNA lacking a binding site) over 2 nM Cy5-labelled DNA.

## RNA-Seq library preparation
RNA was isolated from human iPSC lines, hESCs, HFibs and mESCs using RNeasy Mini Kit (Qiagen), and quality assessed in a 2200 TapeStation System (Agilent) using a high Sensitivity RNA ScreenTape. The concentration of RNA was assessed using a NanoDrop ND-1000 (Marshall Scientific). RNA (RIN > 9.5) libraries were prepared from 500 ng starting material and amplified for 8 cycles using NEBNext Poly(A) mRNA Magnetic Isolation Module (NEB E7490), NEBNext Ultra II RNA Library Prep Kit for Illumina (NEB E7770) and NEBNext Multiplex Oligos for Illumina (NEB E7600) according to the manufacturer's instructions. Equimolar amounts of all libraries were pooled together and sequenced in a NovaSeq 6000 (Edinburgh Genomics facilities) with an SP flow cell, generating 50 bp pair-end reads.

## Bioinformatics
**RNA-seq data analysis.** Uniquely mapped reads were aligned to the Homo sapiens (human) genome assembly GRCh37 (hg19) or the Mus musculus genome assembly MGSCv37 (mm9) using STAR v2.5.3a (--outFilterMultimapNmax 1)[115] and assigned to genes using feature-Counts (Subread v1.5.2)[116]. Read quality was assessed using FastQC (v0.11.9) to confirm high-quality sequences for subsequent analysis, with no need for pre-trimming adaptors or low-quality reads.

For human RNA-seq, reads were normalised to read depth using the median-of-ratios method and rlog transformed (DESeq2)[117] and the top 500 variable genes were used to generate the Pearson correlation plot.

Gene annotation for mES RNA-seq data was performed using featureCounts (subread v2.0.6) to generate a counts table, utilizing the corresponding version of the GTF file for accurate gene annotation. The resulting BAM files from STAR were sorted and indexed using Samtools (v1.9) and converted into bigwig files using DeepTools (v3.5.4 bamCoverage), scaled and normalized to the same library size. Lowly expressed genes were filtered out before normalization and analysis using edgeR (v4.0.16).

For PCA, reads were log-transformed and the prcomp function in R was used to identify the variation between samples and groups. The results were then visualized using the biplot function in R to display the relationships between samples and groups. For differential gene expression analysis, a threshold of log fold change (FC) > 1.5 and false discovery rate (FDR) < 0.05 was applied to identify significant differential expressed genes (DEGs).

**ATAC-seq data analysis.** To assess the chromatin accessibility of mES cells, ATAC-sequencing was performed according to the Buenrostro et al protocol[118]. Briefly, appropriate digitonin titration for cell lysis was tested and determined as 0.006%. Next, 4 × 10^5 cells per mES line were counted, pelleted (4 °C, 3 min, 500 × g), washed once with 1 ml ice-cold ATAC-RSB buffer (10 mM Tris-HCl pH 7.4, 10 mM NaCl, 3 mM MgCl2) and resuspended in 50 μl ice-cold ATAC lysis buffer (ATAC-RSB buffer containing: 0.1% NP40, 0.1% Tween-20 and 0.006% Digitonin). Samples

were incubated on ice (3 min) before adding 1 ml ATAC wash buffer (ATAC-RSB buffer, 0.1% Tween-20) and splitting the sample into 4 replicates (262.5 μl each). Samples were centrifuged (4 °C, 10 min, 500 × g) to pellet the nuclei, which were resuspended in 50 μ freshly prepared ATAC transposition mix (25 μl 2x Illumina Tagment DNA buffer (#20034197), 2.5 μl Illumina Tn5 Transposase (#20034197), 16.5 μl PBS, 0.5 μl 10% Tween-20, 0.5 μl 0.6% digitonin, 5 μl nuclease-free water) before incubation (37 °C, 30 min, 1krpm). Next, the tagmented DNA was purified using the Zymo DNA Clean & Concentrator-5 kit (Zymo Research) following the manufacturer's protocol, eluting in 22 μl EDTA-free elution buffer. To prepare ATAC-seq libraries, purified DNA (20 μl) was amplified by PCR using custom-made indexing primers and the NEBNext High Fidelity 2X Master Mix (1x) for a number of cycles specified by qPCR (8-10) as described previously[119]. Double-sided clean-up was performed using NEB Next Sample Purification Beads (New England Biolabs), adding 0.5x volume to remove >1 kb and 1.3x volume to remove < 150 bp fragments. The quality of the libraries was assessed by the Agilent High Sensitivity D1000 ScreenTape gels. Equimolar amounts of all libraries were pooled together and sequenced in a NovaSeq 6000 (Edinburgh Genomics facilities) with an SP flow cell, generating 50 bp pair-end reads.

**Mapping ChIP-seq reads.** The quality of the next generation sequencing (NGS) raw data (FASTQ files) was measured by FASTQC tool, which all scored QC values of more than 30 across the sequenced 50 bp passing the quality control standard for NGS data. The pair-end sequences were mapped to the Homo sapiens (human) genome assembly GRCh37 (hg19) or the Mus musculus genome assembly MGSCv37 (mm9) using Bowtie (version 2.3.4.1) and very sensitive parameters[120]. Duplicated reads were then removed using MarkDuplicates from the Picard toolkit (http://broadinstitute.github.io/picard/). The sequencing coverage and the insert size distribution were measured from the resulting bam files using Qualimap (version 2.2.1)[121]. The peaks of OCT4 wildtype and mutants as well as SOX2, KLF4 and cMYC showing significant enrichment over input DNA were called using MACS2 (version 2.1.1.20160309)[122] and a fragment size of 200 bp and were controlled to q-value (minimum FDR) cut-off of 0.01. The peaks that overlapped with the ENCODE blacklist were removed.

**Read density heatmaps.** The aligned reads (bam files) were normalised for sequencing coverage to 1x genome depth (reads per genome coverage, RPGC) using the bamCoverage tool from DeepTools and a bin size of 10 bp and ExtendReads parameters[123]. Reads within the ENCODE blacklist were removed from the analysis. The resulting bigwig files were converted to wig format using the bigwigtowig tool[124], which were then converted to bed files using the wigtobed tool[125]. To sort open from closed chromatin peaks, the normalized ATAC-seq tag counts were quantified within the central 300 bp region of each identified peak using the bedmap tool[125]. The peaks were then sorted with an ascending rank-order for ATAC-seq read density. Peaks (central 300 bp) with ATAC-seq densities of 20 RPGC or more were considered open and the rest (less than 20 RPGC) were considered closed. To generate the read density heatmaps, we first computed a density matrix using the DeepTools2 tool computeMatrix reference-point and the following parameters; --referencePoint center, --binsize 10, -b 1000 -a 1000, --sortregions keep, and --averageTypeBins sum using the peak bed files as reference files (-R) and the normalised ChIP-seq and ATAC-seq bigwig files as score files (-S)[123]. The analysis excluded the ENCODE blacklist. We then used the resulting matrix files to generate the heatmaps and profile plots using the DeepTools2 tool plotHeatmap[123].

**Genomic intervals intersection.** Overlapping peaks were isolated if their peak summits were within 100 bp or less from each other using the bedtools window tool (version 2.18)[126].

**Motif discovery.** De Novo motif analysis was carried out using the MEME suite installed in a local LINUX server[127]. First, the DNA sequences (FASTA) were extracted from the central 200 bp of the ChIP-seq peak regions using the BedTools getfasta tool[126]. To use as background, DNA sequences (200 bp) were extracted from genomic regions located 1 Kb upstream from the summit of each peak. All regions were filtered through the ENCODE blacklist. Second, the 1st-order Markov background model was generated using the fasta-get-markov tool. Finally, meme-chip was run using the Fasta sequence files and the corresponding Markov model and the following parameters; -nmeme 600, -meme-mod zoops, -meme-minw 6, -meme-maxw 18, -meme-maxsize 50000000, -dreme-e 0.00001, -dreme-m 20 using the JASPAR core motif database[128]. The most enriched de novo motifs discovered by DREME were analysed by CentriMo to confirm their central enrichment over the background sequences and compared to the canonical motifs using Tomtom. Motif central enrichment were carried out using the CentriMo tool[129].

**Gene ontology analysis.** ChIP-seq peaks were associated with genes using GREAT (Genomic Regions Enrichment of Annotations Tool)[130]. GO enrichment analysis of RNA-seq data was carried out using the PANTHER classification system[131] and DAVID[132,133].

**Proteomic data analysis**
**Identified proteins filter vs control.** All datasets were filtered against their respective controls to filter non-specific interactors using the Perseus Software[134]. 'Reverse', 'Only identified by site' and contaminant proteins were first removed from all datasets. Proteins with at least two peptides were considered. LFQ values were analysed after transformed to log2(x) and missing values were imputed using the minimum value setting. To identify proteins enriched against the control a t-test statistical analysis was performed, using FDR of 0.1- and 1.5-fold change as cut-off values.

**Differential enrichment analysis hESCs vs OSKM 48 h.** Differential enrichment between hES and OSKM 48 h was determined using the Perseus software[134] taking into account the LFQ values from the control-filtered datasets and using a two sided t-test with 250 randomizations, a FDR of 0.1- and 1.5-fold change as the cut-off values. Volcano plot was generated by plotting the fold change value in the x- and the −log t-test p-value for the y-axis. Raw data is shown in Supplementary Data 5.

**Protein network.** The hESCs and OSKM 48 h network was generated with Cytoscape[135] using the STRING app[66] to define interactions based in the Homo sapiens database establishing a confidence (score) cut-off of 0.7. DyNet Analyser app[136] for Cytoscape was used to overlap and visualise both networks. Nodes colouring represents the fold change values defined in the differential enrichment analysis with Perseus.

**PCA and heatmap for ChIP-SICAP hES and OKSM (WT and mutants).** Hierarchical clustering heatmap and PCA plot for ChIP-SICAP analysis between hES and OSKM 48h (WT and mutants) were performed using the R[137] package DEP (Differential Enrichment analysis of Proteomic Data)[138]. Differential enrichment analysis was based on linear models and empirical Bayes statistics with an adjusted p-value of 0.1- and 1.5-fold change as the cut-off values. Normalised LFQ intensity and fold changes are shown in Supplementary Data 6.

**SILAC analysis.** SILAC dataset was analysed with Perseus Software[134]. 'Reverse', 'Only identified by site' and contaminant proteins were first removed. Proteins with at least two peptides and present in at least two replicates were considered. HFibs/hESCs ratios were analysed with a one-sample t-test with a p-value < 0.05 and twofold change as cut-off values. Volcano plot was generated by plotting the fold change value in

the x- and the −log $t$-test $p$-value for the $y$-axis. SILAC and RNA levels heatmaps were generated using the R[137] package *gplots* and the function *heatmap.2*[139].

**Protein functional enrichment.** Gene Ontologies, pathways and complexes enrichment were calculated using the Panther classification system[140] and ConcensusPathDB[141].

**Protein structure.** Human OCT4 3D structure was predicted by AlphaFold2[61]. Human OCT4 DBD was aligned to the mouse OCT4-DBD (3L1P)[35], using the PyMOL Molecular Graphics System, Version 3.0.3, Schrödinger, LLC.

**Protein sequence analysis.** Amino acid sequence alignment and conservation was carried out by Clustal Omega[142]. Protein disorder predictions were calculated using PONDR[143,144] and AlphFold2[61]. MoRFs were predicted using MoRFpred[60]. Prediction of Protein Binding Regions using ANCHOR web server[62,145].

**Reporting summary**

Further information on research design is available in the Nature Portfolio Reporting Summary linked to this article.

## Data availability

All next-generation sequencing data generated as part of this study have been deposited in the Gene Expression Omnibus (GEO) under following accession numbers: GSE287493, GSE286895, GSE287206, GSE286894, GSE286923, GSE287492, GSE287494. Previously published ChIP-seq, ATAC-seq and MNase-seq data used in this study is available under the following GEO accession numbers: GSE167632, GSE168142, GSE168141 (ref. 44), GSE201852 (ref. 65), and GSE120131 (ref. 146). The mass spectrometry proteomics data have been deposited to the ProteomeXchange Consortium via the PRIDE (ref. 147) partner repository with the dataset identifier PXD067538. Unique reagents such as OCT4 mutants are available from the authors. All sequencing data were aligned to the mouse reference genome MGSCv37 (mm9) (PRJNA20689) or the human genome assembly GRCh37 (hg19) (PRJNA438682). Source data are provided with this paper.

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

## Acknowledgements

We thank Profs Katrin Ottersbach, Ian Chambers and Donal O'Carroll for their insightful comments on the manuscript. We thank Prof S. Rosser, the Edinburgh Genome Foundry and the UK Centre for Mammalian Synthetic Biology for support with generating synthetic DNA constructs. We thank Edinburgh Genomics for the sequencing service. Edinburgh Genomics is partly supported through core grants from NERC (R8/H10/56), MRC (MR/K001744/1) and BBSRC (BB/J004243/1). We thank Sukanija Thillainadarasan for assistance with making the OCT4 alanine-stretch mutant constructs. The work was supported by an MRC career development award (MR/N024028/1), CRUK-OHSU Project Award (C65925/A26986), and by the UK Research Councils' Synthetic Biology for Growth programme and of the BBSRC, EPSRC and MRC (BB/M0180404/1) to A.S., B.O., E.H-P., and G.A.R. M.R.A. was funded by a CONACYT Mexico PhD scholarship. M.R.O. is supported by a Ph.D scholarship from the Darwin Trust of Edinburgh. K.F. was supported by the UK Centre for Mammalian Synthetic Biology Edinburgh and CRUK Program Grant (DRCNPG-Nov21\100002). A.A. is funded by a PhD scholarship (1078107040) from King Abdulaziz City for Science and Technology. M.H. is supported by EPSRC DTP PhD.

## Author contributions

A.S. conceived the study and designed the experiments with support from B.O. and M.R.A. B.O. constructed the OCT4 deletion library and performed the mouse iPSCs reprogramming screen, supported by M.L.H. with human iPSC reprogramming experiments. E.H-P. carried out molecular cloning, MEF generation, and pluripotency rescue screen with help from B.O. M.R.A. conducted ChIP-SICAP and proteomic analysis with help from C.S. Luciferase assays were conducted by B.O. with support from E.H-P. and H.N. Immunostaining and imaging were carried out by B.O., M.R.A., E.H-P., A.A., M.N., M.B., and S.Y. Recombinant protein purification, EMSA and nucleosome reconstitution were carried out by G.A.R. RNA-seq of human ESCs and iPSCs were carried out by B.O. ATAC-seq and knock-down experiments during human reprogramming were performed by B.O. Knock-in mouse ES lines were generated by B.O. with support from M.H., S.Y., M.R.O. and K.F. ChIP-seq and RNA-seq in mESCs were conducted by A.A. with support from S.Y. ATAC-seq in mESCs was carried out by M.H. and K.F. ChIP-seq, RNA-seq, ATAC-seq and MNase-seq bioinformatic analysis were carried out by M.R.A., B.O., A.A., M.R.O. and A.S. CRISP-cas9 knock out during reprogramming, in MEFs and in ESCs were performed by M.R.A and M.B. and supported by K.K. Reprogramming to human iTSCs was carried out by M.N. and teratomas were generated by H.Y., who were supervised by Y.B. Blastocysts and chimeras were generated by M.R.P.M and supervised by S.L. A.S. prepared the manuscript with contributions from B.O. and M.R.A., and edited by K.K., H.N., Y.B., and S.L.

## Competing interests

The authors declare no competing interests.
