## [Peer Review file · Nature Communications]

Cell-type-specific functionality encoded within the intrinsically disordered regions of OCT4

Corresponding Author: Dr Abdenour Soufi

Version 0:

Reviewer comments:

Reviewer #1

(Remarks to the Author)

Soufi and colleagues provide a detailed dissection of OCT4 functional domains, investigating their roles in self-renewal, reprogramming, and developmental progression. OCT4 is a central transcription factor required for maintaining pluripotency and inducing reprogramming, and it is also known to influence differentiation. In this study, the authors identify specific regions of OCT4 that are required for distinct biological functions and carry out an extensive characterization of their activities. The work is carefully executed, and the data are both compelling and informative. I have just a few comments:

Figure 1: When stating that a particular OCT4 variant is required for reprogramming but not for ESC self-renewal, it would be important to include qRT-PCR data showing the expression levels of the different OCT4 variants. While the transgene may support self-renewal, it is important to confirm whether it is being expressed at significantly higher levels, which could be compensating for reduced functional activity.

The statement, "OCT4 SLiPERs are crucial for reprogramming but can be dispensable for establishing and maintaining pluripotency throughout early gastrulation," requires clarification. What exactly is meant by "dispensable for establishing pluripotency"? To my knowledge, this has not been clearly demonstrated, and the data presented do not appear to support this specific conclusion. It would be helpful if the authors could clarify or rephrase this statement to better reflect the evidence shown.

Reviewer #2

(Remarks to the Author)

The authors have addressed my comments and this manuscript is now more focused. They have provide considerable amounts of new data and removed some of the more preliminary experiments. In particular the uncoupling of generic transcriptional activation and reprogramming really adds to the paper. I am a little concerned with the late stage chimera data as it is difficult to discern how clonal propagation in different mutant backgrounds influences the ability of these lines to participate in development, and therefore the conclusion that the SLiPERs are required for later development are premature in the absence of rescue of their mutant lines. However, I think that following editorial revision of the text, this paper should be published in Nature Communications.

Reviewers' comments:

Reviewer #1 (Remarks to the Author):

The study by Soufi and colleagues dissects the role of different domains of Oct4 for reprogramming and self-renewal activity. The authors claim that the NT domain of Oct4 has a major role in reprogramming but not in self-renewal. The authors used both mouse and human reprogramming systems to support these claims. The authors then explored possible mechanistic reasons for the “essential” role of this domain in reprogramming and claim that this may be due to a lack of interaction with a subset of identified Oct4 chromatin-proteome. The OCT4 chromatin-proteome on in ESCs and somatic cells is the most interesting aspect of the study but this identified hundreds of factors and the relevance of these long lists are not much explored. As for the manuscript as a whole I have a key issue that fundamentally affects the interpretation of the data as indicated below:

We would like to thank the reviewer for their time and effort in reviewing our manuscript. We greatly appreciate the interest shown in the OCT4 chromatin protein interactome, which we have explored in greater detail in this revision, as explained below. Additionally, we have clarified the main objective of this study, which is to uncover the molecular features that allow OCT4 to perform cell-type-specific functions across various cellular environments.

The key assumption of the study and from which all interpretations are based upon is that reprogramming requires an OCT4 domain(s) not involved in ESC self-renewal. The inference of this “essential domain for reprogramming” is based on the reduced reprogramming ability of Oct4 transgenes displaying mutations in the NT of Oct4, which were still competent to confer self-renewal ability to Oct4 deficient ESCs. However, a simpler and more attractive explanation to such observations is that Oct4 NT mutants have reduced transactivation activity and that the observed differences between reprogramming and self-renewal assays are simply due to their differences in terms of Oct4 activity sensitivity. Self-renewal of pluripotent cells, but not reprogramming, may be compatible with lower Oct4 transactivation capacity and activity which would render the comparison of the two systems as reported by the authors unsuitable. When looking at the self-renewal data in Figure 2 the use of NT Oct4 mutants (Del 10-13 equivalent to Del29-42) actually show lower self-renewal capacity. This is further suggestive that the activity of Oct4 was hindered by the NT mutations and that a much higher dose of Oct4 NT mutant may need to be delivered, relative to Oct4 WT transgene, to drive reprogramming. Another argument that such mutations reduce Oct4 activity but not completely hide it is the fact that the NT region of Oct4 is not conserved across mammalian species.

We have addressed the primary concern raised by the reviewer regarding potential differences in OCT4-NT function during reprogramming compared to its role in the self-renewal of ES cells. This concern suggested that the observed differences might be attributed to the general transactivation function of OCT4, which is more crucial during reprogramming than during ES cell self-renewal. To clarify this issue, we provide further evidence in support of our original conclusions:

1. We have generated eleven additional OCT4 mutants to investigate various essential and non-essential reprogramming domains, which we now refer to as SLiPERs and nonSLiPERs, respectively. These mutants encompass both the OCT4-NT and CT domains (see Fig. 1i). As shown in Fig. 1j and 1k, the removal or substitution of the OCT4-SLiPER that spans amino acids 311-321 (located in the OCT4-CT domain) has little effect on ES cell self-renewal but completely inhibits reprogramming. In contrast, the OCT4-lin-mini mutant, which replaces three non-SLiPERs with linkers, exhibits reduced self-renewal compared to the OCT4-del29-42 (a SLiPER mutant) but still maintains full reprogramming capability. In summary, the relationship between the reprogramming function and ESC self-renewal of OCT4-SLiPERs and nonSLiPERs is not positively correlated, indicating that reprogramming is not more sensitive to OCT4 transactivation than ESC self-renewal.
2. We have directly measured the transactivation capabilities of both OCT4 SLiPER and non-SLiPER mutants using a Luciferase reporter assay (see Fig. 3c). In two separate luciferase reporter constructs driven by the *FGF4* and *UTF1* enhancers, the reprogramming-deficient OCT4-lin29-42 mutant demonstrated strong luciferase transactivation that was comparable to both OCT4-WT and the reprogramming-efficient OCT4-lin95-117 mutant. In contrast, the reprogramming-efficient OCT4-lin-mini mutant exhibited reduced luciferase transactivation. These results suggest that the reprogramming function of OCT4 is not merely associated with a generic transactivation ability. This supports our original conclusion that OCT4-SLiPERs have a specific role in the reprogramming process.
3. We have also investigated whether fusing the transactivation domain of VP16 (VP16-TAD) to OCT4 can substitute OCT4-SLiPERs in reprogramming. Rather than enhancing reprogramming, VP16-TAD reduced the ability of OCT4 to induce pluripotency, as seen previously¹. Importantly, OCT4-lin29-42-VP16-TAD fusion remained deficient in reprogramming like the

non-fusion OCT4-lin29-42 counterpart (Extended data Fig. 4b,c), confirming that OCT4-SLiPERs cannot be replaced by generic transactivation domains in reprogramming.

4. The transactivation of OCT4 has been associated with liquid-liquid phase separation (LLPS) due to the intrinsically disordered characteristics of the OCT4 N-terminal (NT) and C-terminal (CT) regions². Therefore, we investigated the roles of SLiPER and non-SLiPER in OCT4's ability to form protein condensates in early reprogramming. As shown in Figures 3a,b and Extended Data Figure 4a, both mutants of OCT4 with SLiPERs and non-SLiPERs display a similar tendency to form protein condensates during reprogramming. This indicates that the function of OCT4-SLiPERs in reprogramming is not mediated by generic transactivation or LLPS.
5. Importantly, we have demonstrated that the only way to restore the reprogramming activity of the OCT4-SLiPER mutant is by fusing OCT4 with TNIP2. TNIP2 is one of the chromatin-associated partners of OCT4 that we identified through ChIP-SiCAP during reprogramming, but not during ESC self-renewal (Fig. 5g,h). This further supports the idea that OCT4 SLiPERs have a specific role in reprogramming rather than serving as a general transactivation factor.
6. We utilized various computer prediction tools, including AlphaFold2, to discover that OCT4-SLiPERs adopt a quasi-ordered state. These segments may function as molecular recognition features (MoRFs), which are short regions that facilitate interactions with structured proteins and undergo disorder-to-order transitions, also known as induced folding (see Fig. 3a and 3d). This data supports the idea that OCT4-SLiPERs may fold upon specific interactions with protein partners such as TNIP2 during the reprogramming process, but not during self-renewal.

We firmly believe that the new data included in this revised manuscript provide clear evidence that OCT4-SLiPERs play a specific role in the reprogramming process. Therefore, the alternative interpretation—that our reprogramming assay is merely more sensitive to OCT4 transactivation than to ESC self-renewal—is not supported by the data.

The authors also note that in their chromatin-proteome just a few interactors are missing in the category only present in the “essential domain”. How likely is this due to chance? How strong interactors are these? They are not indicated as top rated interactors in figure 5C list. Many hundreds of proteins were identified in the chromatin proteome rendering it somewhat unspecific and likely to generate a few factors absent in the Oct4 NT mutant and present in the other Oct4 transgenes.

To ensure that the proteins identified by ChIP-SiCAP are genuine partners of OCT4 on chromatin, we used the following experimental and data analysis criteria:

1. We employed two controls to eliminate any non-specific interactors: an IgG control in human embryonic stem cells (hESCs) and an OCT4 antibody in human fibroblasts (hFib), which do not express OCT4. As illustrated by the PCA analysis in Extended Data Figure 5c, these controls formed an independent cluster, despite the use of different antibodies and cell types. This indicates the presence of common non-specific contaminants. Moreover, the proteins involved in early reprogramming (OSKM-48h) clustered together, with wild-type (WT) and mutant samples in one group, whereas the ES samples formed a distinct cluster. This high-level analysis reveals that the differences between the OCT4 WT interactors during early reprogramming and t hES cells are more significant than the differences between the OCT4 WT and the mutants in early reprogramming. Notably, this holds true despite a higher number of identified proteins for the OCT4 mutants during early reprogramming. Therefore, it is highly unlikely that the differences in protein partners across the various OCT4 mutants occur by chance.

- We limited our comparison to proteins that were consistently present or absent across both replicates of all OCT4 variants (Fig. 4d). In total, 725 proteins met this stringent criterion. More than 30% of the analysed proteins were found in all OCT4 interactomes (cluster 10), which represented the largest overlap. This was followed by the proteins shared among all OSKM-48h conditions (cluster 2) (Fig. 4d). Notably, the OCT4 partners that were specifically absent during reprogramming in the OCT4-lin29-42 (SLiPER mutant) (shown in cluster 8 of Fig. 4d and illustrated in the figure below) also passed all of the selective criteria, reducing the likelihood of false positive results.
- We have validated cluster 8 proteins functionally and biochemically using independent approaches (Fig. a-e). This confirms that the presence of these proteins is not a result of chance (false positives).
- The most significant evidence we incorporated into the revised manuscript is that TNIP2 can completely restore the reprogramming function of the OCT4-lin29-42 (SLiPER mutant) (see Fig. 5g,h). Through ATAC-seq and knockdown assays, we demonstrate that TNIP2 influences the chromatin accessibility of OCT4 binding sites during the reprogramming process (see Fig. 5i-k).
- In Fig. 4a, we have indicated that TNIP2 is among the top OCT4 interactors (see the figure below for cluster 8 proteins interaction intensity).

Our revised manuscript provides compelling evidence that OCT4 utilizes SLiPERs to specifically interact with certain proteins during the reprogramming process.

In sum this work could be better explained by a reduced transactivation activity of Oct4 NT mutants. This is not novel nor surprising. Conversely, adding further transactivation domains would likely make Oct4 more potent as it would increase Oct4's protein interacting capabilities. Again, this wouldn't be novel either.

Our new data have definitively shown that the reduced transactivation of OCT4-SLiPER mutants cannot solely explain their role in reprogramming. As previously mentioned, adding additional transactivation domains (3x VP16-TADs) inhibits rather than enhances OCT4 reprogramming and cannot rescue the reprogramming activity of the OCT4-SLiPER mutants. In contrast, adding TNIP2 to OCT4, which specifically interacts with OCT4 during reprogramming, can restore the reprogramming activity of the OCT4-SLiPER mutant. Therefore, we believe our conclusions are strongly supported by the presented data. Regarding the novelty of our findings, we define previously unrecognized molecular features of OCT4 within its intrinsic disordered regions (IDRs), which extend beyond the known transactivation and liquid-liquid phase separation (LLPS).

Reviewer #2 (Remarks to the Author):

Since the 1980s it has been clear that DNA binding and transcriptional activation/repression are regulated by separable domains, and that there are domains outside of the DNA binding domain that are required for cooperativity. However, little has been done to link cell fate choice and lineage specification to these specific regulatory domains. Ozkan et al. does just this, making impressive progress linking domain specific interactions to cell type specific reprogramming. By generating a range of Oct4 mutants that span the whole protein, they are able to pinpoint specific regions that are essential for reprogramming, but not for the support of self-renewal in pluripotent cells. Furthermore, they are able to propose a potential mechanism for why the defined segments of Oct4 in the amino terminus are crucial for reprogramming, by coupling ChIP-seq with ChIP-SiCAP to match the binding of Oct4 mutants with binding partners on chromatin. Intriguingly, they find a strong correlation between the capacity of mutants to reprogram and their ability to recognize sites in inaccessible genomic regions including Oct4 targets normally bound in pluripotent cells. Using their gain and loss of function mutants, they identify binding partners that are unique to reprogramming and could be key to regulating Oct4's activity. They identify a set of 7 proteins/interactors that are required for optimal reprogramming. In addition, they test the role of Oct4 amino terminal domains in reprogramming to trophoblast stem cells.

Taken together this paper covers an impressive quantity of work that attempts to provide unique insight into the amino terminal regulatory domain of Oct4 and its function in reprogramming. While the majority of the data in this paper is suitable for publication in Nature Cell Biology, there are a few outstanding issues (general comments 1 & 2) and perhaps the story could be better focused (general comment 3). For example, there is more than enough data in the paper and perhaps the details of the mouse phenotype (general comment 1) are beyond the scope of this manuscript.

We sincerely appreciate the reviewer's positive feedback and insightful comments. As outlined below, we have addressed all of the reviewer's suggestions. We agree with the reviewer regarding the mouse phenotype, which we have completely removed from the revised manuscript. However, we have generated three separate embryonic stem (ES) cell lines containing the essential reprogramming domain mutation, referred to as SLiPER, in the endogenous *Pou1f5* gene. We used these lines to directly assess this mutation's impact on ES cell capacity to differentiate in chimeric models, teratoma assays, and *in vitro* studies. These lines have also helped us to reveal an unexpected role for SLiPER in OCT4 binding within ESCs, which appears to influence their lineage commitment without affecting self-renewal. Interestingly, our new experiments indicate that OCT4's ability to reprogram somatic cells is also linked to its capacity to drive the lineage commitment of ES cells.

General Comments:

1. The phenotype of the Oct4 N-terminal deletion in mice is very preliminary and is not the same mutations used in the reprogramming experiments that have been characterized extensively. The authors wish to interpret this in terms of the capacity of the protein (Oct4) to prime for later gene expression or development. However, there is little analysis of these mice and the notion that Oct4 is required later in development (postnatal) is not assessed. There are varying reports of expression of Oct4 later in development and adult life (see citations below) and the cause of the mortality of these mice could be a result of a highly specific phenotype in any of the organ system, since it is only reported as ability to reach adulthood. Without a more detailed characterization of the phenotype and its underlying molecular basis, the only conclusion that can be made is that mice with a mutation in the N terminal region undergo normal embryonic development, but that the N terminus is required for some aspect of fetal development or postnatal life.

A few examples of reports of somatic Oct4 expression:

OCT4 Spliced Variants Are Differentially Expressed in Human Pluripotent and Nonpluripotent Cells
<https://stemcells.journals.onlinelibrary.wiley.com/doi/full/10.1634/stemcells.2008-0530>

Novel Variants of Oct-3/4 Gene Expressed in Mouse Somatic Cells
<https://www.jbc.org/content/283/45/30997.full>

Expression patterns of ABCG2, Bmi-1, Oct-3/4, and Yap in the developing mouse incisor
<https://www.sciencedirect.com/science/article/pii/S1567133X1000116X?via=ihub#b0110>

BGEM: An In Situ Hybridization Database of Gene Expression in the Embryonic and Adult Mouse Nervous System
<https://journals.plos.org/plosbiology/article?id=10.1371/journal.pbio.0040086>

We acknowledge the reviewer's concern regarding the poorly defined mouse phenotype and have accordingly removed all mouse phenotypic data from the revised manuscript. Instead, we have developed three clonal ES lines in which the OCT4-SLiPER mutation—specifically, the region we characterized in the paper—has been either removed or replaced with a linker within the endogenous *Pou5f1* gene using CRISPR-Cas9 knock-in technology. As shown in Extended Data Figures 7c and 7d, these ESC lines can form teratomas containing derivatives from all three germ layers, indicating their pluripotency. Interestingly, in a more controlled setting, such as chimaera assays, these OCT4-SLiPER mutant ESC cells contribute normally to the blastocyst up to E6.5 but begin to exhibit defects starting at E8.5 (see Figure 6c and 6d). We were unable to generate any chimeric E13.5 embryos from these lines (Figure 6f).

Additionally, the ES mutant lines showed robust self-renewal and resistance to transitioning out of pluripotency when LIF is withdrawn in vitro (Figures 6h and 6i). Through extensive characterisation of these OCT4-SLiPER mutant ES cell lines, we discovered an aberrant pattern of OCT4 binding and deregulated gene expression of lineage specific TFs (Figures 7a-e). We suggest that this may disrupt the timely differentiation of ES cells, which is essential during embryonic development.

Altogether, we believe that OCT4-SLiPERs are crucial for lineage commitment during late gastrulation. However, we agree with the reviewer that this does not exclude the potential function of OCT4-SLiPERs in other somatic cell contexts in adult life, a point we have included in the discussion.

2. The trophoblast reprogramming requires better analysis. While I appreciate that this work is based on a paper in revision, it is hard to understand how Oct4 could mediate TSC reprogramming. Is the exogenous Oct4 still expressed?

The paper describing the reprogramming of human fibroblasts into induced trophoblast stem cells (iTSCs) using OCT4, GATA3, KLF4, and c-MYC (GTKM) has now been published³. We have also included further characterization of human iTSC cells using immunofluorescence (see Fig. 2h). These iTSC cells are fully reprogrammed, and therefore the exogenous OCT4 has been silenced in these cells. Thus, we are measuring the OCT4-SLiPER function during the iTSC reprogramming process.

As Oct4 has been reported to reprogram murine TSCs to iPSCs (<https://www.ncbi.nlm.nih.gov/pmc/articles/PMC3126346/> and <https://stemcellsjournalsonline.wiley.com/doi/full/10.1002/stem.617>) and is not required for iTSC programming of MEFs (<https://pubmed.ncbi.nlm.nih.gov/26412560/>), this is puzzling.

The papers cited by the reviewer describe the reprogramming of TS cells and iTSC cells in the mouse model. However, the role of OCT4 in reprogramming to iTSC cells, which we describe in our paper, is based on the human model. Recent studies have demonstrated that establishing the trophectoderm differs significantly between human and mouse^{4,5}. Specifically, OCT4 play a crucial role in establishing human TS cells, which contrasts the mouse model⁶. Moreover, our recent study clearly indicates that OCT4 can reprogram human somatic cells to iTSC cells independently of pluripotency³. We have previously shown that reprogramming to mouse iTSC cells is achieved using different sets of TFs that do not include OCT4⁷. In summary, in our study we dissect the reprogramming function of OCT4 to human iPSC and iTSC cells apart from each other by simply removing different parts of the protein.

a) While, GRN in human may not be the same as in mouse the authors need to address the differences. Especially since mouse Oct4 is used for the reprogramming of the human fibroblasts and that equivalent mutations in human Oct4 are not produced or tested. How does this relate to role of Oct4 in establishing trophectoderm in early human but not mouse embryos?

It is hard to imagine Oct4 being necessary for trophoblast establishment/propagation in human, when its expression is downregulated in defined trophoblast by day 10.

<http://www.pnas.org/lookup/doi/10.1073/pnas.1911362116>

<https://www.sciencedirect.com/science/article/pii/S0012160612006744?via=ihub>

The role of OCT4 in the establishment of human trophoblasts and the reprogramming to iTSC cells has been extensively researched, as previously described, and is beyond the scope of this study^{3,6}. We have utilized human OCT4 in all of our reprogramming experiments. Additionally, we generated a mouse equivalent of OCT4 mutants when targeting the endogenous Pou5f1 gene in mouse ES cells. We believe that the section on reprogramming to human iTSC cells is a minor part of

our revised manuscript, included solely to demonstrate that OCT4 can employ different domains in various reprogramming contexts. Following the reviewer's recommendation, we have removed all CHIP-SICAP analysis that was not experimentally validated using the iTSC reprogramming model.

b) In addition, human TSCs have only recently been identified, but appear different from their murine counter parts in key respects.

Recent transcriptomes suggest they represent 1st trimester placenta.

<https://www.sciencedirect.com/science/article/pii/S1934590917304563?via=ihub>

<https://elifesciences.org/articles/52504>

While we appreciate the sensitivity of another paper currently being in revision. Perhaps the authors could use the transcriptomes in the above papers as a means to bench mark and demonstrate that they have genuine human iTSCs.

Our research paper on human iTSC cell reprogramming has been published, including comprehensive transcriptomic and chromatin characterization³. We have also characterized clonal iTSC cell lines generated from our OCT4 mutants, as shown in Fig. 2h, using OCT4 wildtype as a control.

c) Finally, why was the analysis used to identify potential Oct4 binders during iTSC reprogramming done on the ChIP-seq and ChIP-SICAP from iPSC reprogramming only. It would be much more convincing if the analysis was repeated for iTSC reprogramming.

We agree with the reviewer that those analyses may not be relevant to iTSC cell reprogramming. Consequently, we have removed them from the revised manuscript.

3. The fact that the authors have identified 7 factors that potentially facilitate the ability of Oct4 to recognize its sites buried in chromatin is a fundamental observation that could perhaps be strengthened. Some assessment of how these factors affect the access of Oct4 to inaccessible sites would strengthen this manuscript, in addition to an enhanced discussion of potential mechanism.

a) Performing Oct4 ChIP-seq in the knockout of these 7 factors.

We have now added extensive characterization and molecular mechanism of TNIP2, which is one of the 7 factors we identified that interact with OCT4 by CHIP-SICAP. We concentrated on TNIP2 as is the only factor that is translocated from the cytoplasm to chromatin and is uniquely required during reprogramming (Fig. 5e and Extended data Fig. 6). We have now added three sets of experiments to strengthen our conclusions:

- 1- We have mapped that TNIP2-CT can readily translocate to the nucleus. We have also shown that TNIP2CT-OCT4 fusion is located in the nucleus during reprogramming (fig. 5f).
- 2- Remarkably, we show that fusing TNIP2-CT to OCT4-SLiPER mutant can fully restore its reprogramming function in contrast to overexpressing TNIP2-CT as a separate molecule (Fig. 5g,h). This demonstrates that the physical association of TNIP2-CT with OCT4 is important for reprogramming. Importantly, fusing VP16-TAD to OCT4 could not rescue the reprogramming function of OCT4-SLiPER mutant (extended data Fig. 4b,c)

3- We have carried out ATAC-seq after knocking down TNIP2 (Fig. i,j). This has shown a significant decrease in chromatin opening of the closed sites targeted by OCT4 during reprogramming (Fig. 5k).

Altogether, we have provided further evidence supporting our conclusion that OCT4 employs SLiPERs to interact with a specific set of proteins that are required for reprogramming but not ES cell self-renewal. We have also illustrated a possible mechanism explaining how OCT4 partners can help open up chromatin during reprogramming.

b) The authors hypothesize a link between reprogramming specific Oct4 targets and apoptotic/stress response. To test this they could compare ROS and cleaved caspases levels during reprogramming and pluripotency maintenance in the presence/absence of these factors that are reprogramming specific.

We conducted a Caspase assay to demonstrate that the levels of apoptosis induced by OCT4 during early reprogramming are significantly increased by the OCT4-SLiPER mutant compared to the wild type and other mutants (Fig. 3h). However, we did not observe any increase in apoptosis when culturing ES cells with the OCT4-SLiPER mutation under LIF and LIF+2i conditions. Additionally, gene expression analysis using RNA sequencing has not revealed any differences in apoptosis between the ES cell mutants and the wild type, despite the presence of aberrant gene expression and OCT4 mutant binding in the mutant ES cells. Therefore, we conclude that the OCT4-SLiPER mutant specifically enhances apoptosis during the reprogramming process.

Conclusion:

It is our opinion that if the authors want to maintain the contention that Oct4 is required for lineage priming (point 1) or necessary for iTSC formation (point 2) then further data is required. If removed, then authors could focus on essential OCT4 interactors (point 3) and clarify the writing. There are also a few detailed points that should be addressed.

We appreciate the reviewer's insightful comments. We believe we have effectively addressed all three points raised, and our revised manuscript is now significantly improved and more focused.

Specific Comments:

1. Clarification on Figure5: A supplemental table showing the filtering process (removal of non-specific interactors) for the ChIP-SiCAP data should be included.

Supplementary Table 7 has been added with more details on the filtering process in the material and method section.

In Figure 5d the authors perform a t-test (visualized as a volcano plot) comparing OSKM 48hrs to hESCs when each only has two replicates, a minimum of three replicates is recommended for t-tests. Additionally, they use an FDR of 0.1, which is larger than the traditional 0.01 (1%FDR).

We apologize for the mistake, the FDR value is 0.01 not 0.1. We conducted two biological replicates, but each contained three technical replicates. We believe that this is sufficient to obtain statistically significant data.

Finally, the color designations on the volcano plot are unclear. How are the “enriched uniquely” and “shared but more enriched” groups separated? Does unique mean it was not identified in the either replicate of the other condition? This could be easily shown with a Venn diagram. Are the “shared but more enriched” proteins included in with the string networks of Figure 5e-g?

Correct; the “shared but more enriched” are proteins identified in both conditions but more enriched in one condition. The “enriched uniquely” are only identified in one condition. We have now provided more explanation in the materials and methods. In extended data Figure 5d we have included all three categories with the string networks, which are colour-coded to indicate enrichment.

How many proteins are in the three categories of the SILAC heatmaps (Figure 5h should include “n” number) and how were these categories determined? Is it also based on the volcano plot?

The ‘n’ value is now presented in Figure 4c. The categories were defined based on t-test analysis, with a p-value of less than 0.05 and a fold change greater than 2. More details have been included in the materials and methods section, as well as in the figure legend. Additionally, Supplementary Table 8, which contains all SILAC proteins, has been added.

2. There are several phosphorylation or ubiquitin sites reported near or in the deletion mutant ranges a.a. 95-117 (S102, S106, K118) a.a. 125-138 (K137) and a.a. 341-354 (T343, S347). How do these sites relate to the alanine mutations performed?

S102, S106, and K118 (S105, S107, and K123 in humans), and K137 (K140 in humans) have not been investigated by alanine substitution. T343, S347 (T351/352, S349 in human) show no impact in reprogramming (see ala-116 in extended data Fig. 1a). We haven’t specifically investigated phosphorylation and ubiquitination in this manuscript, so we cannot rule out that they play distinct functions during reprogramming and ESC self-renewal.

Reviewer #3 (Remarks to the Author):

In this manuscript, the authors systematically dissected the functions of different domains of OCT4 protein in reprogramming, pluripotency maintenance and lineage commitment. The results suggest that different OCT4 domains have differential roles in reprogramming and cell fate control. This study provides a novel insight into the mechanism by which OCT4 regulates cell fate during cellular reprogramming and embryonic development. Overall the study was well designed, the data are convincing, and the conclusions are appropriate. The flow of the paper can be improved by removing some non-essential data.

We thank the reviewer for their positive feedback and support.

Minor points:

1. The authors need to provide more detailed statistical methods for Fig 6d, Extended Data Figs. 1c and 1g.

More details on the statistical analysis has been provided in the revised manuscript.

2. The authors need to provide scale bars for Figs 1c, 7C, Extended Data Figs 1e, 1f, 5f, 6f, 7a, 7c, and 7d. What are the values of the scale bars in Fig. 3f?

All scale bars are now included in the figure and described in the figure legends.

3. Many of the error bars in Extended Data Fig 1g are big, indicating the data may not be reliable. The authors need to repeat these experiments.

More repeats have been carried out (n=5). However, it is important to note that spontaneous differentiation can be highly variable, which is reflected in the large error bars. Thus, this is inherent to the nature of these experiments.

4. The authors need to confirm whether OCT4 Δ 29-42 and Δ 95-117 are expressed in cells after transfection.

The expression of all OCT4 variants has been measured at the mRNA and protein levels (Extended data Figure 2e and f).

5. In Fig. 6e, the bands for OCT4 WT, lin29-42, and lin95-117 do not seem to correspond to their molecular weights.

The expected molecular weight (MW) of human OCT4 is 38571 Da. However, it appears to run slightly higher than anticipated in western blots, which is consistent with the information provided on the commercial antibody website <<https://www.abcam.com/en-us/products/primary-antibodies/oct4-antibody-epr17929-chip-grade-ab181557#>>. This discrepancy may be attributed to post-translational modifications or the presence of SDS-resistant secondary structures of OCT4. Moreover, the other OCT4 variants run somewhat faster than the wild type (WT), consistent with their smaller size. Importantly, this antibody specifically recognises OCT4, appearing as a single band in western blots (as illustrated in figure 5e and extended data figure 2f). These sizes are also equivalent to the purified recombinant OCT4 counterparts (Extended data figure 4e).

6. Using the del-72 iPS cells, the authors confirmed that the flexibility of the linker and not the rigid helix of OCT4 is essential for inducing pluripotency. A previous study suggested that the linker region of OCT4 is a.a. 76-92 (Nature Cell Biology 15:295-301, 2013). Please confirm whether 72 amino acid is in the linker domain for OCT4.

We have confirmed that del-72 is located within the rigid helix of the OCT4-DBD linker region (extended data figure 1d). To support our findings, we utilised AlphaFold2 to generate the structure of human OCT4-DBD and aligned it with the previously published crystal structure of mouse OCT4-DBD published by the Schöler lab⁸. We have provided the alignment between the human and mouse OCT4-DBD linker to eliminate any potential confusion arising from the difference in amino acid numbering of the two species. The numbering used by Schöler's lab (76-92) is for the mouse OCT4-DBD only, while we are using the full-length human OCT4 (extended data figure 1d).

References:

1. Hammachi, F., Morrison, G.M., Sharov, A.A., Livigni, A., Narayan, S., Papapetrou, E.P., O'Malley, J., Kaji, K., Ko, M.S., Ptashne, M., and Brickman, J.M. (2012). Transcriptional activation by Oct4 is sufficient for the maintenance and induction of pluripotency. *Cell Rep* *1*, 99-109. [10.1016/j.celrep.2011.12.002](https://doi.org/10.1016/j.celrep.2011.12.002).
2. Boija, A., Klein, I.A., Sabari, B.R., Dall'Agnesse, A., Coffey, E.L., Zamudio, A.V., Li, C.H., Shrinivas, K., Manteiga, J.C., Hannett, N.M., et al. (2018). Transcription Factors Activate Genes through the Phase-Separation Capacity of Their Activation Domains. *Cell* *175*, 1842-1855 e1816. [10.1016/j.cell.2018.10.042](https://doi.org/10.1016/j.cell.2018.10.042).
3. Naama, M., Rahamim, M., Zayat, V., Sebban, S., Radwan, A., Orzech, D., Lasry, R., Ifrah, A., Jaber, M., Sabag, O., et al. (2023). Pluripotency-independent induction of human trophoblast stem cells from fibroblasts. *Nat Commun* *14*, 3359. [10.1038/s41467-023-39104-1](https://doi.org/10.1038/s41467-023-39104-1).
4. Blakeley, P., Fogarty, N.M.E., del Valle, I., Wamaitha, S.E., Hu, T.X., Elder, K., Snell, P., Christie, L., Robson, P., and Niakan, K.K. (2015). Defining the three cell lineages of the human blastocyst by single-cell RNA-seq. *Development* *142*, 3151-3165. [10.1242/dev.123547](https://doi.org/10.1242/dev.123547).
5. Petropoulos, S., Edsgård, D., Reinius, B., Deng, Q., Panula, S.P., Codeluppi, S., Reyes, A.P., Linnarsson, S., Sandberg, R., and Lanner, F. (2016). Single-Cell RNA-Seq Reveals Lineage and X Chromosome Dynamics in Human Preimplantation Embryos. *Cell* *167*, 285. <https://doi.org/10.1016/j.cell.2016.08.009>.
6. Fogarty, N.M.E., McCarthy, A., Snijders, K.E., Powell, B.E., Kubikova, N., Blakeley, P., Lea, R., Elder, K., Wamaitha, S.E., Kim, D., et al. (2017). Genome editing reveals a role for OCT4 in human embryogenesis. *Nature advance online publication*. [10.1038/nature24033](https://doi.org/10.1038/nature24033)
<http://www.nature.com/nature/journal/vaop/ncurrent/abs/nature24033.html#supplementary-information>.
7. Benchetrit, H., Herman, S., van Wietmarschen, N., Wu, T., Makedonski, K., Maoz, N., Yom Tov, N., Stave, D., Lasry, R., Zayat, V., et al. (2015). Extensive Nuclear Reprogramming Underlies Lineage Conversion into Functional Trophoblast Stem-like Cells. *Cell Stem Cell* *17*, 543-556. [10.1016/j.stem.2015.08.006](https://doi.org/10.1016/j.stem.2015.08.006).
8. Esch, D., Vahokoski, J., Groves, M.R., Pogenberg, V., Cojocar, V., Vom Bruch, H., Han, D., Drexler, H.C., Arauzo-Bravo, M.J., Ng, C.K., et al. (2013). A unique Oct4 interface is crucial for reprogramming to pluripotency. *Nat Cell Biol* *15*, 295-301. [10.1038/ncb2680](https://doi.org/10.1038/ncb2680).

>Reviewer #1 (Remarks to the Author):

>Soufi and colleagues provide a detailed dissection of OCT4 functional domains,
>investigating their roles in self-renewal, reprogramming, and developmental progression.
>OCT4 is a central transcription factor required for maintaining pluripotency and inducing
>reprogramming, and it is also known to influence differentiation. In this study, the authors
>identify specific regions of OCT4 that are required for distinct biological functions and carry
>out an extensive characterization of their activities. The work is carefully executed, and the
>data are both compelling and informative. I have just a few comments:

We thank the reviewer for their positive and insightful comments.

>Figure 1: When stating that a particular OCT4 variant is required for reprogramming but not
>for ESC self-renewal, it would be important to include qRT-PCR data showing the
>expression levels of the different OCT4 variants. While the transgene may support self-
>renewal, it is important to confirm whether it is being expressed at significantly higher
>levels, which could be compensating for reduced functional activity.

We have now measured OCT4 expression during ESC self-renewal rescue of several mutants compared to WT using qPCR (Supplementary Fig. 3c), showing that the expression levels of OCT4 mutants do not correlate with their ability to support ESC self-renewal. This is consistent in all mutants with exception of OCT4 delta-mini mutant, which showed low expression levels and impaired self-renewal ability. We have also measured the expression levels of OCT4 mutants compared to WT during reprogramming by qPCR and Western blots

(Supplementary Figs. 1c, 3a and 3b). Again, the expression levels of OCT4 mutants don't correlate with their reprogramming activity.

>The statement, "OCT4 SLiPERs are crucial for reprogramming but can be dispensable for establishing and maintaining pluripotency throughout early gastrulation," requires clarification. What exactly is meant by "dispensable for establishing pluripotency"? To my knowledge, this has not been clearly demonstrated, and the data presented do not appear to support this specific conclusion. It would be helpful if the authors could clarify or rephrase this statement to better reflect the evidence shown.

We have changed that statement in the discussion to: "OCT4 SLiPERs are crucial for reprogramming but can be dispensable for supporting ESC self-renewal". We have also changed similar statements in the abstract.

>Reviewer #2 (Remarks to the Author):

>The authors have addressed my comments and this manuscript is now more focused. They have provide considerable amounts of new data and removed some of the more preliminary experiments. In particular the uncoupling of generic transcriptional activation and reprogramming really adds to the paper. I am a little concerned with the late stage chimera data as it is difficult to discern how clonal propagation in different mutant backgrounds influences the ability of these lines to participate in development, and therefore the conclusion that the SLiPERs are required for later development are premature in the absence of rescue of their mutant lines. However, I think that following editorial revision of the text, this paper should be published in Nature Communications.

We thank the reviewer for their time and support. We agree with the reviewer that a rescue experiment will further support the conclusion that SLiPERs play important role later in development. We have tried extensively to introduce TNIP2 into the OCT4 locus to see if it can rescue ESC differentiation and development, as it did in reprogramming. However, we couldn't obtain any clones with the correct TNIP2 insertion after screening hundreds. Therefore, we have made this clearer by adding the following statement to the discussion: "However, it's important to confirm the role of OCT4-SLiPERs in development. For example, rescue experiments using OCT4-SLiPER mutant fused to TNIP2, similar to what we observed in reprogramming, could provide further evidence."

Please find below a summary of our manuscript as requested in the editorial check list:

"Systematic dissection of OCT4 reveals how intrinsically disordered regions can be used to serve specific functions during reprogramming and embryonic development. This can be exploited to engineer more efficient reprogramming factors".

Please don't hesitate to contact me if you need any further information.